# Interpretable and Generalizable Graph Neural Networks via Subgraph Multilinear Extension

## Abstract

Interpretable graph neural networks (XGNNs) are widely adopted in scientific applications involving graph-structured data. Previous approaches predominantly adopt the attention-based mechanism to learn edge or node importance for extracting and making predictions with the interpretable subgraph. However, the representational properties and limitations of these methods remain inadequately explored. In this work, we present a theoretical framework that formulates interpretable subgraph learning with the multilinear extension of the subgraph distribution, which we term as *subgraph multilinear extension* (`SubMT`). Extracting the desired interpretable subgraph requires an accurate approximation of `SubMT`, yet we find that the existing XGNNs can have a huge gap in fitting `SubMT`. Consequently, the `SubMT` approximation failure will lead to the degenerated interpretability of the extracted subgraphs. To mitigate the issue, we design a new XGNN architecture called **G**raph **M**ultilinear ne**T** (`GMT`), which is provably more powerful in approximating `SubMT`. We empirically validate our theoretical findings on a number of graph classification benchmarks. The results demonstrate that `GMT` outperforms the state-of-the-art up to $10\%$ in terms of both interpretability and generalizability measures.

## 1 Introduction

Graph Neural Networks (GNNs) have been widely used in scientific applications (Wang et al., 2023; Zhang et al., 2023) such as Physics (Bapst et al., 2020), Chemistry (Gilmer et al., 2017; Jumper et al., 2021), Quantum mechanics (Kochkov et al., 2021), Materials (Schütt et al., 2017) and Cosmology (Villanueva-Domingo et al., 2021). In pursuit of scientific discoveries, it often requires GNNs to be able to generalize to *unseen or Out-of-Distribution* (OOD) graphs (Gui et al., 2022; Ji et al., 2022; Zhang et al., 2023), and also provide *interpretations* of the predictions that are crucial for scientists to collect insights (Xie & Grossman, 2017; Cranmer et al., 2020; Dai et al., 2021) and promote better scientific practice (Murray & Rees, 2009; Wencel-Delord & Glorius, 2013). Recently there has been a surge of interest in developing intrinsically interpretable and generalizable GNNs (XGNNs) (Yu et al., 2021; Miao et al., 2022; Wu et al., 2022b; Chen et al., 2022a; Miao et al., 2023). In contrast to *post-hoc* explanation approaches (Ying et al., 2019; Yuan et al., 2020a; Vu & Thai, 2020; Luo et al., 2020; Yuan et al., 2021; Lin et al., 2021; 2022a) which is shown to be suboptimal in interpretation and sensitive to pre-trained GNNs performance (Miao et al., 2022; 2023), XGNNs are able to provide both reliable explanations and (OOD) generalizable predictions under the proper guidance such as information bottleneck (Yu et al., 2021) and causality (Chen et al., 2022a).

Indeed, the faithful interpretation and the reliable generalization are the *two sides of the same coin* for XGNNs. Grounded in the causal assumptions of data generation processes (Wu et al., 2022a; Miao et al., 2022; Wu et al., 2022b), XGNNs assume that there exists a causal subgraph which holds a causal relation with the target label. Predictions made solely based on the causal subgraph are generalizable under various graph distribution shifts (Chen et al., 2022a). Therefore, XGNNs typically adopt the two-step paradigm that first extracts a subgraph of the input graphs and then predicts the labels. To circumvent the inherent discreteness of subgraphs, XGNNs often learn the sampling probability for each edge or node with the attention mechanism and extract the subgraph with high attention scores (Miao et al., 2022). Predictions are then made via a weighted message passing scheme with the attention scores (Wu et al., 2022b; Chen et al., 2022a). Despite the notable

**Subgraph Extraction as Subgraph Multilinear Extension**  **Subgraph Classification**

$G \sim \mathcal{D}_{tr}$       $\widehat{G}_c = \mathbb{E}_{G_c \overset{g}{\sim} G}[G_c]$       $f_c(\widehat{G}_c) = f_c(\mathbb{E}_{G_c \overset{g}{\sim} G}[G_c])$

Figure 1: Illustration of Subgraph Multilinear Extension (SubMT). The task is to classify whether a graph contains a specific "house" or "cycle" motif. An XGNN $f = f_c \circ g$ predicts the label with the classifier $f_c$ based on the extracted soft subgraph $\widehat{G}_c = g(G)$, denoted as the central graph. Different depths of edge colors refer to the sampling probability of the edge. $\widehat{G}_c = g(G)$ corresponds to a subgraph distribution where the sampling probability for each subgraph $G_c$ (i.e., subgraphs with solid lines in the figure). SubMT extends GNNs to accept soft subgraph inputs by estimating the subgraph conditional prediction as the expectation of each possible subgraph $\mathbb{E}[f_c(G_c)]$. Interpretable subgraph learning requires accurate estimation of SubMT. Yet existing XGNNs that directly take the soft subgraph $\widehat{G}_c$ as classifier GNN inputs fail to reliably estimate SubMT. In contrast, GMT aims to bridge the gap between SubMT and $f_c(\widehat{G}_c)$ by learning a neural SubMT to align with SubMT.

success of the paradigm in enhancing both interpretability and out-of-distribution (OOD) (Miao et al., 2022; 2023; Chen et al., 2022a), there is little theoretical understanding of the representational properties and limitations of XGNNs, and whether they can provide faithful interpretations.

In this work, we present a framework to analyze the expressiveness and evaluate the faithfulness of XGNNs. Our framework is inspired by the close connection between interpretable subgraph learning and multilinear extension, a powerful tool for solving classical combinatorial optimization problems (Călinescu et al., 2007). In fact, the subgraph learning in XGNNs naturally resembles the multilinear extension of the subgraph predictivity, which we termed as *subgraph multilinear extension* (SubMT). Extracting the truth interpretable subgraph requires a precise approximation of SubMT. However, we show that the prevalent attention-based paradigm can fail to reliably approximate SubMT (Sec. 3.2). Consequently, the SubMT approximation failure will decrease the interpretability of the subgraph for predicting the target label. More specifically, we instantiate the issue via a causal framework and propose a novel interpretability measure called *counterfactual fidelity*, i.e., the sensitivity of the prediction with respect to small perturbations to the extracted subgraphs (Sec. 4.2). Although faithful interpretation should have a high counterfactual fidelity with the prediction, we find that XGNNs implemented with the prevalent paradigm only have a low counterfactual fidelity.

Aiming to bridge the gap, we propose a simple yet effective XGNN architecture called **G**raph **M**ultilinear ne**T** (GMT). The core design of GMT is inspired by the SubMT formulation, which performs random subgraph sampling to reduce the SubMT approximation error. We prove that GMT is provably more powerful in approximating SubMT (Sec. 5). We validate our theoretical findings through extensive experiments on multiple graph classification benchmarks. The results demonstrate that GMT improves the state-of-the-art up to $10\%$ in both interpretability and generalizability (Sec. 6).

## 2 PRELIMINARIES AND RELATED WORK

We begin by introducing preliminary concepts of XGNNs and leave more details to Appendix B.1. For ease of understanding, a table of the notations for key concepts is given in Appendix A.

**Interpretable GNNs.** Let $G = (A, X)$ be a graph with node set $V = \{v_1, v_2, ..., v_n\}$ and edge set $E = \{e_1, e_2, ..., e_m\}$, where $A \in \{0, 1\}^{n \times n}$ is the adjacency matrix and $X \in \mathbb{R}^{n \times d}$ is the node feature matrix. In this work, we focus on interpretable GNNs (or XGNNs) for the graph classification

task, while the results can be generalized to node-level tasks as well (Wu et al., 2020). Given each sample from training data $\mathcal{D}_{\mathrm{tr}} = (G^i, Y^i)$, an interpretable GNN $f := h \circ g$ aims to identify a (causal) subgraph $G_c \subseteq G$ via a subgraph extractor GNN $g : \mathcal{G} \to \mathcal{G}_c$, and then predicts the label via a subgraph classifier GNN $f_c : \mathcal{G}_c \to \mathcal{Y}$, where $\mathcal{G}, \mathcal{G}_c, \mathcal{Y}$ are the spaces of graphs, subgraphs, and the labels, respectively (Yu et al., 2021). Although *post-hoc* explanation approaches also aim to find an interpretable subgraph as the explanation for the model prediction (Ying et al., 2019; Yuan et al., 2020a; Vu & Thai, 2020; Luo et al., 2020; Yuan et al., 2021; Lin et al., 2021; 2022a), they are shown to be suboptimal in interpretation performance and sensitive to the performance of the pre-trained GNNs (Miao et al., 2022). Therefore, this work focuses on *intrinsic interpretable* GNNs (XGNNs).

A predominant approach to implement XGNNs is to incorporate the idea of information bottleneck (Tishby et al., 1999), such that $G_c$ keeps the minimal sufficient information of $G$ about $Y$ (Yu et al., 2021; 2022; Miao et al., 2022; 2023; Yang et al., 2023), which can be formulated as

$$\max_{G_c} I(G_c; Y) - \lambda I(G_c; G), \ G_c \sim g(G), \tag{1}$$

where the maximizing $I(G_c; Y)$ endows the interpretability of $G_c$ while minimizing $I(G_c; G)$ ensures $G_c$ captures only the most necessary information, $\lambda$ is a hyperparamter trade off between the two objectives. In addition to minimizing $I(G_c; G)$, there are also alternative approaches that impose different constraints such as causal invariance (Chen et al., 2022a; Li et al., 2022) or disentanglement (Wu et al., 2022b; Sui et al., 2022; Liu et al., 2022a; Fan et al., 2022) to identify the desired subgraphs. When extracting the subgraph, XGNNs adopts the attention mechanism to learn the sampling probability of each edge or node, which avoids the complicated Monte Carlo tree search used in other alternative implementations (Zhang et al., 2022). Specifically, given node representation learned by message passing $H_i \in \mathbb{R}^h$ for each node $i$, XGNNs either learns a **node attention** $\alpha_i \in \mathbb{R}_+ = \sigma(a(H_i))$ via the attention function $a : \mathbb{R}^h \to \mathbb{R}_+$, or the **edge attention** $\alpha_e \in \mathbb{R}_+ = \sigma(a([H_u, H_v]))$ for each edge $e = (u, v)$ via the attention function $a : \mathbb{R}^{2h} \to \mathbb{R}_+$, where $\sigma(\cdot)$ is a sigmoid function. $\boldsymbol{\alpha} = [\alpha_1, ..., \alpha_m]^T$ essentially elicits a subgraph distribution of the interpretable subgraph. In this work, we focus on edge-centric subgraph sampling as it is most widely used in XGNNs while our method can be easily generalized to node-centric approaches.

**Faithful interpretation and (OOD) generalization.** The faithfulness of interpretation is critical to all interpretable and explainable methods (Ribeiro et al., 2016; Lipton, 2018; Alvarez-Melis & Jaakkola, 2018; Jain & Wallace, 2019). There are several metrics developed to measure the faithfulness of graph explanations, such as fidelity (Yuan et al., 2020b; Amara et al., 2022), counterfactual robustness (Bajaj et al., 2021; Prado-Romero et al., 2022; Ma et al., 2022), and equivalence (Crabbé & van der Schaar, 2023), which are however limited to post-hoc graph explanation methods. In contrast, we develop the first faithfulness measure for XGNNs in terms of counterfactual invariance.

In fact, the generalization ability and the faithfulness of the interpretation are naturally intertwined in XGNNs. XGNNs need to extract the underlying ground-truth subgraph in order to make correct predictions on unseen graphs (Miao et al., 2022). When distribution shifts are present during testing, the underlying subgraph that has a causal relationship with the target label (or causal subgraphs) naturally becomes the ground-truth subgraph that needs to be learned by XGNNs (Chen et al., 2022a).

**Multilinear extension** serves as a powerful tool for maximizing combinatorial functions, especially for submodular set function maximization (Călinescu et al., 2007; Vondrak, 2008; Bian et al., 2019; Sahin et al., 2020; Karalias et al., 2022). It is the expected value of a set function under the fully factorized Bernoulli distribution. In this work, we are the first to identify subgraph multilinear extension as the factorized subgraph distribution for interpretable subgraph learning.

## 3    ON THE EXPRESSIVITY OF INTERPRETABLE GNNS

In this section, we present our theoretical framework for characterizing the expressivity of XGNNs. Since all of the existing approaches need to maximize $I(G_c; Y)$ regardless of the regularization on $G_c$, we focus on the modeling of the subgraph distribution that maximizes $I(G_c; Y)$.

### 3.1    SUBGRAPH MULTILINEAR EXTENSION.

The need for maximizing $I(G_c; Y)$ originates from extracting information in $G$ to predict $Y$ with $f_c$,

$$\arg\max_{f_c} I(G; Y) = \arg\max_{f_c} [H(Y) - H(Y|G)] = \arg\min_{f_c} H(Y|G), \tag{2}$$

where the last equality is due to the irrelevance of $H(Y)$ and $f_c$. For each sample $(G, Y)$, XGNN then adopts the subgraph extractor $g$ to extract a subgraph $G_c \sim g(G)$, and take $G_c$ as the input of $f_c$ to predict $Y$. Then, Eq. 2 is realized as follows[1]: let $L(\cdot)$ be the cross-entropy loss, then

$$\arg\min_{g, f_c} \mathbb{E}_{(G, Y) \sim \mathcal{D}_{\text{tr}}}[-\log P(Y | \mathbb{E}_{G_c \overset{g}{\sim} G} G_c)] = \mathbb{E}_{(G, Y) \sim \mathcal{D}_{\text{tr}}}[L(f_c(\boldsymbol{\alpha}; G), Y)], \quad (3)$$

where $\boldsymbol{\alpha} \in \mathbb{R}_+^m$ is the attention score elicited from the subgraph extractor $g$. We leave more details about the deduction of Eq. 3 in Appendix B.2. Note that $f_c$ is a GNN defined only for *discrete* graph-structured inputs (i.e., $\boldsymbol{\alpha} \in \{0, 1\}^m$), while Eq. 3 imposes continuous inputs to $f_c$. Considering $f_c(G_c)$ is a *set function* with respect to node/edge index subsets of $G$ (i.e., subgraphs $G_c$), and the parameterization of $P(G)$ in XGNNs (Miao et al., 2022), we resort to the *multilinear extension* of $f_c(G_c)$. Multilinear extension for set functions has been extensively studied in the domain of solving classical combinatorial optimization problems (Călinescu et al., 2007; Karalias et al., 2022).

**Definition 3.1** (Subgraph multilinear extension (`SubMT`)). *Given the attention score $\boldsymbol{\alpha} \in [0, 1]^m$ as sampling probability of $G_c$, XGNNs factorize $P(G)$ as independent Bernoulli distributions on edges:*

$$P(G_c | G) = \prod_{e \in G_c} \alpha_e \prod_{e \in G/G_c} (1 - \alpha_e),$$

*which elicits the multilinear extension of $f_c(G_c)$ in Eq. 3 as:*

$$F_c(\boldsymbol{\alpha}; G) := \sum_{G_c \in G} f_c(G_c) \prod_{e \in G_c} \alpha_e \prod_{e \in G/G_c} (1 - \alpha_e) = \mathbb{E}_{G_c \overset{g}{\sim} G} f_c(G_c). \quad (4)$$

The parameterization of $P(G)$ is widely employed in XGNNs (Miao et al., 2022; Chen et al., 2022a), which implicitly assumes the random graph data model (Erdos & Rényi, 1984). Def. 3.1 can also be generalized to other graph models with the corresponding parameterization of $P(G)$ (Snijders & Nowicki, 1997; Lovász & Szegedy, 2006). When a XGNN approximates `SubMT` well, we have:

**Definition 3.2** ($\epsilon$-`SubMT` approximation). *Let $d(\cdot, \cdot)$ be a distribution distance metric, a XGNN $f = f_c \circ g$ $\epsilon$-approximates `SubMT` (Def. 3.1), if there exists $\epsilon \in \mathbb{R}_+$ such that $d(P_f(Y|G), P(Y|G)) \leq \epsilon$ where $P(Y|G) \in \mathbb{R}^{|\mathcal{Y}|}$ is the ground truth conditional label distribution, and $P_f(Y|G) \in \mathbb{R}^{|\mathcal{Y}|}$ is the predicted label distribution for $G$ via a XGNN $f$, i.e., $P_f(Y|G) = f_c(\mathbb{E}_{G_c \overset{g}{\sim} G} G_c)$.*

Def. 3.2 is a natural requirement for XGNN that approximates `SubMT` properly. With the definition of `SubMT`, we can write the problem of Eq. 3 as the following:

$$\arg\min_{g, f_c} \mathbb{E}_{(G, Y) \sim \mathcal{D}_{\text{tr}}}[L(\mathbb{E}_{G_c \overset{g}{\sim} G} f_c(G_c), Y)] = \mathbb{E}_{(G, Y) \sim \mathcal{D}_{\text{tr}}} L(F_c(\boldsymbol{\alpha}; G), Y), \quad (5)$$

Intuitively, optimizing for $g, f_c$ in Eq. 3 requires an accurate estimation of `SubMT`.

## 3.2 ISSUES OF EXISTING APPROACHES

In general, evaluating `SubMT` requires $\mathcal{O}(2^m)$ calls of Eq. 4. However, existing XGNNs take a "shortcut" and introduce a soft subgraph $\widehat{G}_c$ with the adjacency matrix as the attention matrix $\widehat{A}$ where $A_{u,v} = \alpha_e, \forall e = (u, v) \in E$, to estimate Eq. 3 via weighted message passing (Miao et al., 2022):

$$\arg\min_{g, f_c} \mathbb{E}_{(G, Y) \sim \mathcal{D}_{\text{tr}}}[L(\mathbb{E}_{G_c \overset{g}{\sim} G} f_c(G_c), Y)] = \mathbb{E}_{(G, Y) \sim \mathcal{D}_{\text{tr}}}[L(f_c(\widehat{G}_c), Y)]. \quad (6)$$

From the edge-centric perspective, the introduction of $\widehat{G}_c$ seems to be natural at first glance, as:

$$\widehat{G}_c = \mathbb{E}_{G_c \overset{g}{\sim} G} G_c = (X, \widehat{A}). \quad (7)$$

However, Eq. 6 holds only when $f_c$ is *linear*. More formally, as $X$ is fixed, for the sake of brevity, let $f_c(A) := \mathbb{E}_{G_c \overset{g}{\sim} G}[f_c((X, A))]$, then Eq. 6 requires the following to be hold:

$$f_c(\widehat{A}) = f_c(\mathbb{E}[A]) = \mathbb{E}[f_c(A)], \quad (8)$$

where the first equality is by the definition of $\widehat{A}$, while the last equality adheres to the equality of Eq. 6. Obviously $f_c(\cdot)$ is a non-linear function even with a linearized GNN (Wu et al., 2019) with linear activations and pooling such as sum pooling, which can be written as:

$$f_c(G_c) = \rho(\widehat{A}^k X \boldsymbol{W}), \quad (9)$$

where $\rho$ is the pooling, $k$ is the number of layers and $\boldsymbol{W} \in \mathbb{R}^{h \times h}$ are the learnable weights. Therefore,

---

[1] With a bit of abuse of notations, we will omit the unnecessary superscript of samples for the sake of clarity.

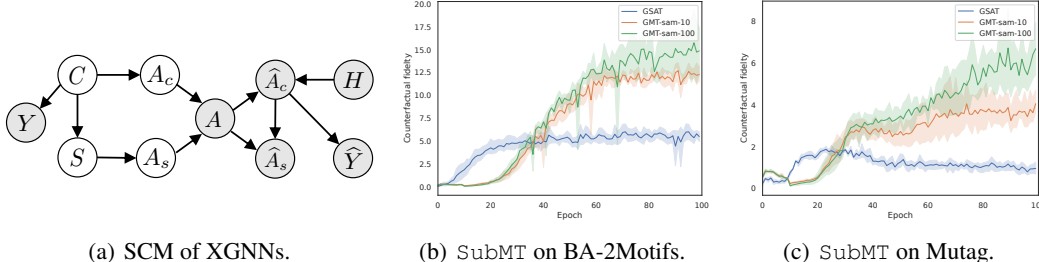

(a) SCM of XGNNs.      (b) `SubMT` on BA-2Motifs.      (c) `SubMT` on Mutag.

Figure 2: Illustration of counterfactual faithfulness.

**Proposition 3.3.** *Eq. 8 with linear GNNs (Eq. 9) and $k > 1$ can not approximate* `SubMT` *(Def. 3.1).*

The proof is given in Appendix D.2. Empirical verifications are also provided in Appendix F.6.. For example, when $k = 2$ and $|\mathcal{Y}| = 1$, Eq. 9 is convex, and we have $f_c(\mathbb{E}[A]) \leq \mathbb{E}[f_c(A)]$ due to Jensen's inequality, which introduces the Jensen gap as $\mathbb{E}[f_c(A)] - f_c(\widehat{A})$ when fitting `SubMT`.

## 4 ON THE GENERALIZATION AND INTERPRETABILITY: A CAUSAL VIEW

To further understand the consequences of the `SubMT` approximation issue, we conduct a causal analysis of the interpretation faithfulness in XGNNs. Without loss of generality, our discussion focuses on the edge-centric view of data generation and interpretation.

### 4.1 CAUSAL MODEL OF INTERPRETABLE GNNs

**Data generation.** We consider the same data model as previous works (Bevilacqua et al., 2021; Miao et al., 2022; Chen et al., 2022a), where the underlying causal subgraph $G_c$ and the spurious subgraph $G_s$ will be assembled via some underlying assembling process. As we focus on the edge-centric view, our following discussion will focus on the graph structures $A_c$ and $A_s$ of the subgraphs. Full details of the structural causal model are deferred to Appendix C.1.

As shown in Fig. 2(a), there are latent causal and spurious variables $C$ and $S$ that have the invariant and spurious correlations with the label $Y$ across different training and test distributions, respectively. $C$ and $S$ correspondingly control the generation of the graph structure of the causal subgraph $G_c$, and the spurious subgraph $G_s$. For example, when generating $A_c$ and $A_s$, $C$ and $S$ will specify the number of nodes in $A_c$ and $A_s$ and also the edge sampling probability for edges in $A_c$ and $A_s$.

**Interpretation.** Correspondingly, XGNNs uses a subgraph extractor to predict the causal and spurious subgraphs $\widehat{G}_c$ and $\widehat{G}_s$, respectively. The extraction aims to reverse the generation and recover the structure of the underlying causal subgraph $A_c$. We denote the XGNN architecture and the hyperparameter settings as $H$. $H$ takes $A$ as inputs to learn the edge sampling probability via the attention mechanism and then obtain $\widehat{A}_c$. Once $\widehat{A}_c$ is determined, $\widehat{A}_s = \mathbf{1} - \widehat{A}_c$ is also obtained by taking the complementary part. Then, the extracted causal and spurious subgraphs are obtained with $\widehat{G}_c = (X, \widehat{A}_c)$ and $\widehat{G}_s = (X, \widehat{A}_s)$, respectively. The classifier then uses $\widehat{G}_c$ to make the prediction $\widehat{Y}$.

### 4.2 CAUSAL FAITHFULNESS OF XGNNs

With the aforementioned causal model, we are able to specify the causal desiderata for faithful XGNNs. When a XGNN fails to accurately approximate `SubMT`, the estimated label conditional probability will have a huge gap from the ground truth. The failure will bias the optimization of the subgraph extractor $g$ and lead to the degenerated interpretability of $\widehat{A}$. More concretely, the recovery of $\widehat{A}$ to the underlying $A$ will be worse, which further affects the extraction of $G_c$ and brings both worse interpretation and (OOD) generalization performance. As a single measure such as the interpretation or generalization may not fully reflect the consequence or even exhibit conflicted information[2], we consider a direct notion that jointly consider the interpretability and generalizabiliy to measure the causal faithfulness of XGNNs, inspired by Jain & Wallace (2019).

---

[2]For example, in the experiments of Miao et al. (2022), higher interpretation performance does not necessarily correlate with higher generalization performance.

**Definition 4.1** (($\delta, \epsilon$)-counterfactual fidelity). *Given a meaningful minimal distance $\delta > 0$, let $d(\cdot, \cdot)$ be a distribution distance metric , if a XGNN $f = f_c \circ g$ commits to the $\epsilon$−counterfactual fidelity, then there exist $\epsilon > 0$ such that, $\forall G, \widetilde{G}$ that $d(P(Y|G), P(Y|\widetilde{G})) \geq \delta$, the following holds:*

$$d(P_f(Y|\widetilde{G}), P_f(Y|G)) \geq \epsilon\delta.$$

Intuitively, if the extracted interpretable subgraph $\widehat{G}_c$ is faithful to the target label, then the predictions made based on $\widehat{G}_c$ are sensitive to any perturbations on $\widehat{G}_c$. Different from counterfactual interpretability (Prado-Romero et al., 2022; Guo et al., 2023) that seeks minimum modifications to change the predictions, ($\delta, \epsilon$)-counterfactual fidelity measures how sensitive are the predictions to the changes of the interpretable subgraphs. A higher fidelity implies better interpretability and is also a natural behavior of a XGNN that approximates `SubMT` well.

**Proposition 4.2.** *If a XGNN $f$ $\epsilon$-approximates `SubMT`, $f$ satisfies $(\delta, 1 - \frac{2\epsilon}{\delta})$-counterfactual fidelity.*

The proof is given in Appendix D.3. Intuitively, Proposition 4.2 implies that the counterfactual fidelity is an effective measure for the approximation ability of `SubMT` in terms of Def. 3.2.

**Practical estimation of counterfactual fidelity.** Since it is hard to enumerate every possible $\widetilde{G}$, to verify Def. 4.1, we consider a random attention matrix $\widetilde{A} \sim \sigma(\mathcal{N}(\mu_{\widehat{H}_A}, \sigma_{\widehat{H}_A}))$, where $\mu_{\widehat{H}_A}$ and $\sigma_{\widehat{H}_A}$ are the mean and standard deviation of the pre-attention matrix $\widehat{H}_A$ (The adjacency matrix with the unnormalized attention). Each non-symmetric entry in $\widetilde{A}$ is sampled independently following the factorization of $P(G)$. We randomly sample $\widetilde{A}$ by $k$ times and calculate the following:

$$c_{\widehat{G}_c} = \frac{1}{k} \sum_{i=1}^{k} d(f_c(Y|\widetilde{G}_c^i), f_c(Y|\widehat{G}_c)), \tag{10}$$

where $\widetilde{G}_c^i = (X, \widetilde{A}_c^i)$ and $d$ is total variation distance. We compute $c_{\widehat{G}_c}$ for the state-of-the-art XGNN `GSAT` (Miao et al., 2022). Shown as in Fig. 2(b), 2(c), we plot the counterfactual fidelity of `GSAT` on BA-2Motifs and Mutag datasets against is 2 to 3 times lower than the simulated `SubMT` with 10 and 100 sampling rounds. We provide a more detailed discussion in Appendix C.2 and Appendix F.5.

## 5 BUILDING RELIABLE INTERPRETABLE AND GENERALIZABLE GNNS

The aforementioned gap motivates us to propose a new XGNN architecture, called **G**raph **M**ultilinear ne**T** (GMT), to provide both faithful interpretability and reliable (OOD) generalizability. GMT have two variants, i.e., `GMT-lin` and `GMT-sam`, motivated by resolving the failures mentioned in Sec. 3.2.

### 5.1 LINEARIZED GRAPH MULTILINEAR NETWORK

Note that the main reason for the failure of Eq. 8 is because of the non-linearity of the expectation to the $k$ weighted message passing with $k > 1$. If $k$ can be reduced to $1$, then the linearity can be preserved to ensure a better approximation of `SubMT`. More formally, for a XGNN $f$ with linearized GNN as the classifier, if $\exists T \in \mathbb{R}^{d \times d}$ such that $T \cdot f_c(G_c) = P(Y|G_c)$ ($f_c$ is linear), then, let

(GMT-lin) $$f_c^l(G_c) = \rho(\widehat{A} \odot A^{k-1} X \boldsymbol{W}), \tag{11}$$

we can incorporate `GMT-lin` into Eq. 8 and have the following holds:

$$f_c^l(\widehat{G}_c) = \boldsymbol{T} \cdot f_c(\widehat{G}_c) = \mathbb{E}[f_c(G_c)],$$

due to the linearity of $f_c^l(G_c)$ with respect to $G_c$ (i.e., $A$). During training, $\boldsymbol{T}$ can be further absorbed into $\boldsymbol{W}$, which implies `GMT-lin` is able to fit to `SubMT`. Compared to the previous weighted message passing scheme with linearized GNN (Eq. 9), `GMT-lin` improves the linearity by reducing the number of weighted message passing rounds to 1. We show the simple strategy can already achieve better interpretability than the state-of-the-art methods even with non-linear GNNs in experiments.

### 5.2 GRAPH MULTILINEAR NETWORK WITH RANDOM SUBGRAPH SAMPLING

Although `GMT-lin` works for linearized GNNs, the non-linear GNNs are more widely used in practice (Xu et al., 2019), where `GMT-lin` may also suffer from the `SubMT` approximation failure.

To overcome the issue, inspired by the SubMT formulation, we propose a random subgraph sampling approach, that performs Markov Chain Monte Carlo (MCMC) sampling to approximate SubMT. More concretely, given the attention matrix $\widehat{A}$, we perform $t$ rounds of random subgraph sampling from the subgraph distribution elicited by $\widehat{A}$ (or equivalently $\widehat{G}_c = (X, \widehat{A})$ as in SubMT (Def. 3.1), and obtain $t$ i.i.d. random subgraph samples $\{G_c^i\}_{i=1}^{t}$ for estimating SubMT as the following

$$(\text{GMT-sam}) \qquad f_c^s(\widehat{G}_c) = \frac{1}{t} \sum_{i=1}^{t} f_c(Y|G_c^i), \qquad (12)$$

where $f_c$ is the classifier GNN that takes discrete subgraphs as inputs.

**Theorem 5.1.** *Given the attention matrix $\widehat{A}$, and the distribution distance metric $d$ as the total variation distance, let $C = |\mathcal{Y}|$, for a GMT-sam with $t$ i.i.d. samples of $G_c^i \sim P(G_c|G)$, then, there exists $\epsilon \in \mathbb{R}_+$ such that, with a probability at least $1 - e^{-t\epsilon^2/4}$, GMT-sam $\frac{\epsilon C}{2}$-approximates SubMT and satisfies $(\delta, 1 - \frac{\epsilon C}{\delta})$ counterfactual fidelity.*

The proof for Theorem 5.1 is given in Appendix D.4. Intuitively, with more random subgraph samples drawn from $P(G_c|G)$, GMT-sam obtains a more accurate estimation of SubMT. However, it will incur more practical challenges such as the a) gradient of discrete sampling and b) computational overhead. To overcome the challenges a) and b), we incorporate the following two techniques.

**Backpropagation of discrete sampling.** To enable gradient backpropagation with the sampled subgraphs, we also incorporate gradient estimation techniques such as Gumbel softmax and straight-through estimator (Jang et al., 2017; Maddison et al., 2017). Compared to the state-of-the-art XGNN GSAT (Miao et al., 2022), this scheme brings two additional benefits: (i) reduces the gradient biases in discrete sampling with Gumbel softmax; (ii) avoids weighted message passing and alleviates the input distribution gap to the graph encoder when shared in both $f_c$ and $g$ as in GSAT.

**Learning neural subgraph multilinear extension.** Although GMT trained with GMT-sam improve interpretability, GMT-sam still requires multiple random subgraph sampling to approximate SubMT and costs much additional overhead. To this end, we propose to learn a neural SubMT that only requires single sampling, based on the trained subgraph extractor $g$ with GMT-sam.

Learning the neural SubMT is essentially to approximate the MCMC with a neural network, though it is inherently challenging to approximate MCMC (Johndrow et al., 2020; Papamarkou et al., 2022). Nevertheless, the feasibility of neural SubMT learning is backed by the inherent causal subgraph assumption of (Chen et al., 2022a), once the causal subgraph is correctly identified, simply learning the statistical correlation between the subgraph and the label is sufficient to recover the causal relation.

Therefore, we propose to simply re-train a new classifier GNN with the frozen subgraph extractor, to distill the knowledge contained in $\widehat{G}_c$ about $Y$. This scheme also brings additional benefits over the originally trained classifier, which focuses on providing the gradient guidance for finding proper $G_c$ instead of learning all the available statistical correlations between $\widehat{G}_c$ and $Y$. More details and discussions on the implementations can be found in Appendix E.

**The number sampling rounds.** Although the estimation of SubMT will be more accurate with the increased sampling rounds, it unnecessarily brings improvements. First, as shown in Fig. 3, the performance may be saturated with moderately sufficient samplings. Besides, the performance may degenerate as more sampling rounds can affect the optimization, as discussed in Appendix E.2

## 6 EXPERIMENTAL EVALUATIONS

We conduct extensive experiments to evaluate GMT with different backbones and on multiple graph classification benchmarks, and compare both the interpretability and (OOD) generalizability with the traditional post-hoc interpretation methods and the state-of-the-art XGNNs. We will briefly introduce the datasets, baselines, and experiment setups, and leave more details in Appendix F.

### 6.1 EXPERIMENTAL SETTINGS

**Datasets.** We consider both the regular and geometric graph classification benchmarks following the XGNN literature (Miao et al., 2022; 2023). For regular graphs, we include BA-2MOTIFS (Luo

Table 1: Interpretation Performance (AUC) on regular graph datasets. The shadowed entries are the results with the mean-1*std larger than the mean of the corresponding best baselines.

| GNN | METHOD | BA-2MOTIFS | MUTAG | MNIST-75SP | SPURIOUS-MOTIF | | |
| | | | | | $b = 0.5$ | $b = 0.7$ | $b = 0.9$ |
| --- | --- | --- | --- | --- | --- | --- | --- |
| | GNNEXPLAINER | $67.35_{\pm 3.29}$ | $61.98_{\pm 5.45}$ | $59.01_{\pm 2.04}$ | $62.62_{\pm 1.35}$ | $62.25_{\pm 3.61}$ | $58.86_{\pm 1.93}$ |
| | PGEXPLAINER | $84.59_{\pm 9.09}$ | $60.91_{\pm 17.10}$ | $69.34_{\pm 4.32}$ | $69.54_{\pm 5.64}$ | $72.33_{\pm 9.18}$ | $72.34_{\pm 2.91}$ |
| GIN | GRAPHMASK | $92.54_{\pm 8.07}$ | $62.23_{\pm 9.01}$ | $73.10_{\pm 6.41}$ | $72.06_{\pm 5.58}$ | $73.06_{\pm 4.91}$ | $66.68_{\pm 6.96}$ |
| | IB-SUBGRAPH | $86.06_{\pm 28.37}$ | $91.04_{\pm 6.59}$ | $51.20_{\pm 5.12}$ | $57.29_{\pm 14.35}$ | $62.89_{\pm 15.59}$ | $47.29_{\pm 13.39}$ |
| | DIR | $82.78_{\pm 10.97}$ | $64.44_{\pm 28.81}$ | $32.35_{\pm 9.39}$ | $78.15_{\pm 1.32}$ | $77.68_{\pm 1.22}$ | $49.08_{\pm 3.66}$ |
| | GSAT | $98.85_{\pm 0.47}$ | $99.35_{\pm 0.95}$ | $80.47_{\pm 1.86}$ | $74.49_{\pm 4.46}$ | $72.95_{\pm 6.40}$ | $65.25_{\pm 4.42}$ |
| GIN | GMT-LIN | $98.36_{\pm 0.56}$ | $99.86_{\pm 0.09}$ | $82.98_{\pm 1.49}$ | $76.06_{\pm 6.39}$ | $76.50_{\pm 5.63}$ | $\mathbf{80.57}_{\pm 2.59}$ |
| | GMT-SAM | $\mathbf{99.62}_{\pm 0.11}$ | $\mathbf{99.87}_{\pm 0.11}$ | $\mathbf{86.50}_{\pm 1.80}$ | $\mathbf{85.50}_{\pm 2.40}$ | $\mathbf{84.67}_{\pm 2.38}$ | $73.49_{\pm 5.33}$ |
| | GSAT | $89.35_{\pm 5.41}$ | $99.00_{\pm 0.37}$ | $85.72_{\pm 1.10}$ | $79.84_{\pm 3.21}$ | $79.76_{\pm 3.66}$ | $80.70_{\pm 5.45}$ |
| PNA | GMT-LIN | $95.79_{\pm 7.30}$ | $99.58_{\pm 0.17}$ | $85.02_{\pm 1.03}$ | $80.19_{\pm 2.22}$ | $84.74_{\pm 1.82}$ | $85.08_{\pm 3.85}$ |
| | GMT-SAM | $\mathbf{99.60}_{\pm 0.48}$ | $\mathbf{99.89}_{\pm 0.05}$ | $\mathbf{87.34}_{\pm 1.79}$ | $\mathbf{88.27}_{\pm 1.71}$ | $\mathbf{86.58}_{\pm 1.89}$ | $\mathbf{85.26}_{\pm 1.92}$ |

Table 2: Prediction Performance (Acc.) on regular graph datasets. The shadowed entries are the results with the mean-1*std larger than the mean of the corresponding best baselines.

| GNN | METHOD | MOLHIV (AUC) | GRAPH-SST2 | MNIST-75SP | SPURIOUS-MOTIF | | |
| | | | | | $b = 0.5$ | $b = 0.7$ | $b = 0.9$ |
| --- | --- | --- | --- | --- | --- | --- | --- |
| | GIN | $76.69_{\pm 1.25}$ | $82.73_{\pm 0.77}$ | $95.74_{\pm 0.36}$ | $39.87_{\pm 1.30}$ | $39.04_{\pm 1.62}$ | $38.57_{\pm 2.31}$ |
| GIN | IB-SUBGRAPH | $76.43_{\pm 2.65}$ | $82.99_{\pm 0.67}$ | $93.10_{\pm 1.32}$ | $54.36_{\pm 7.09}$ | $48.51_{\pm 5.76}$ | $46.19_{\pm 5.63}$ |
| | DIR | $76.34_{\pm 1.01}$ | $82.32_{\pm 0.85}$ | $88.51_{\pm 2.57}$ | $45.49_{\pm 3.81}$ | $41.13_{\pm 2.62}$ | $37.61_{\pm 2.02}$ |
| | GSAT | $76.12_{\pm 0.91}$ | $83.14_{\pm 0.96}$ | $96.20_{\pm 1.48}$ | $47.45_{\pm 5.87}$ | $43.57_{\pm 2.43}$ | $45.39_{\pm 5.02}$ |
| GIN | GMT-LIN | $76.87_{\pm 1.12}$ | $83.19_{\pm 1.28}$ | $96.01_{\pm 0.25}$ | $47.69_{\pm 4.93}$ | $53.11_{\pm 4.12}$ | $46.22_{\pm 4.18}$ |
| | GMT-SAM | $\mathbf{77.22}_{\pm 0.93}$ | $\mathbf{83.62}_{\pm 0.50}$ | $\mathbf{96.50}_{\pm 0.19}$ | $\mathbf{60.09}_{\pm 2.40}$ | $\mathbf{54.34}_{\pm 4.04}$ | $\mathbf{55.83}_{\pm 5.68}$ |
| | PNA (NO SCALARS) | $78.91_{\pm 1.04}$ | $79.87_{\pm 1.02}$ | $87.20_{\pm 5.61}$ | $68.15_{\pm 2.39}$ | $66.35_{\pm 3.34}$ | $61.40_{\pm 3.56}$ |
| PNA | GSAT | $79.82_{\pm 0.67}$ | $80.90_{\pm 0.37}$ | $93.69_{\pm 0.73}$ | $68.41_{\pm 1.76}$ | $67.78_{\pm 3.22}$ | $51.51_{\pm 2.98}$ |
| | GMT-LIN | $80.05_{\pm 0.71}$ | $81.18_{\pm 0.47}$ | $94.44_{\pm 0.49}$ | $69.33_{\pm 1.42}$ | $64.49_{\pm 3.51}$ | $58.30_{\pm 6.61}$ |
| | GMT-SAM | $\mathbf{80.58}_{\pm 0.83}$ | $\mathbf{82.36}_{\pm 0.96}$ | $\mathbf{95.75}_{\pm 0.42}$ | $\mathbf{71.98}_{\pm 3.44}$ | $\mathbf{69.68}_{\pm 3.99}$ | $\mathbf{67.90}_{\pm 3.60}$ |

et al., 2020), MUTAG (Debnath et al., 1991), MNIST-75SP (Knyazev et al., 2019), which are widely evaluated by post-hoc explanation approaches (Yuan et al., 2020b), as well as SPURIOUS-MOTIF (Wu et al., 2022b), GRAPH-SST2 (Socher et al., 2013; Yuan et al., 2020b) and OGBG-MOLHIV (Hu et al., 2020) where there exist various graph distribution shifts. For geometric graphs, we consider ACTSTRACK, TAU3MU, SYNMOL and PLBIND curated by Miao et al. (2023).

**Baselines.** For post-hoc methods, we mainly adopt the results from the previous works (Miao et al., 2022; 2023), including GNNExplainer (Ying et al., 2019), PGExplainer (Luo et al., 2020), GraphMask (Schlichtkrull et al., 2021) for regular graph benchmarks, and BernMask, BernMask-P, that are modified from GNNExplainer and PGExplainer, GradGeo (Shrikumar et al., 2017), and Grad-Cam (Selvaraju et al., 2017) that are extended for geometric data, as well as PointMask (Taghanaki et al., 2020) developed specifically for geometric data. For XGNNs, since we focus on the interpretation performance, we mainly compared with XGNNs that have the state-of-the-art interpretation abilities, i.e., GSAT (Miao et al., 2022) and LRI (Miao et al., 2023), which also have excellent OOD generalization performance than other XGNNs (Gui et al., 2022). We also include two representative XGNNs baselines, DIR (Wu et al., 2022b) and IB-subgraph (Yu et al., 2021) for regular graph data.

**Training and evaluation.** We consider three backbones GIN (Xu et al., 2019) and PNA (Corso et al., 2020) for regular graph data, EGNN (Satorras et al., 2021) for geometric data. All methods adopted the identical graph encoder, and optimization protocol for fair comparisons. We tune the hyperparameters as recommended by previous works. More details are given in Appendix F.2.

## 6.2 EXPERIMENTAL RESULTS AND ANALYSIS

**Interpretation performance.** As shown in Table. 1, compared to post-hoc based methods (in the first row), and GSAT, both GMT-lin and GMT-sam lead to non-trivial improvements for interpretation performance. Especially, in challenging Spurious-Motif datasets where there contain distribution shifts, GMT-sam brings improvements than GSAT up to 15% with GIN, and up to 8% with PNA. In challenging realistic dataset MNIST-75sp, GMT-sam also improves GSAT up to 6%.

**Generalization performance.** Table 2 illustrates the prediction accuracy on regular graph datasets. We again observe consistent improvements for diverse datasets spanning from molecule graphs to

Table 3: Interpretation performance on the geometric learning datasets. The shadowed entries are the results with the mean-1*std larger than the mean of the corresponding best baselines.

| | ACTSTRACK | | TAU3MU | | SYNMOL | | PLBIND | |
|---|---|---|---|---|---|---|---|---|
| | ROC AUC | PREC@12 | ROC AUC | PREC@12 | ROC AUC | PREC@12 | ROC AUC | PREC@12 |
| RANDOM | 50 | 21 | 50 | 35 | 50 | 31 | 50 | 45 |
| GRADGEO | $69.31_{\pm0.89}$ | $33.54_{\pm1.23}$ | $78.04_{\pm0.57}$ | $64.18_{\pm1.25}$ | $76.38_{\pm4.96}$ | $64.72_{\pm3.75}$ | $58.11_{\pm2.91}$ | $64.78_{\pm4.73}$ |
| BERNMASK | $54.23_{\pm4.31}$ | $20.46_{\pm5.46}$ | $71.58_{\pm0.69}$ | $60.51_{\pm0.76}$ | $76.38_{\pm4.96}$ | $64.72_{\pm3.75}$ | $52.23_{\pm4.45}$ | $41.50_{\pm9.77}$ |
| BERNMASK-P | $22.87_{\pm3.33}$ | $11.29_{\pm5.46}$ | $70.72_{\pm5.10}$ | $55.50_{\pm6.26}$ | $87.06_{\pm7.12}$ | $77.11_{\pm7.58}$ | $51.98_{\pm4.66}$ | $59.20_{\pm5.48}$ |
| POINTMASK | $49.20_{\pm1.51}$ | $20.54_{\pm1.71}$ | $55.93_{\pm4.85}$ | $39.65_{\pm7.14}$ | $66.46_{\pm6.86}$ | $53.93_{\pm1.94}$ | $50.00_{\pm0.00}$ | $45.10_{\pm0.00}$ |
| GRADGAM | $75.19_{\pm1.91}$ | $75.94_{\pm2.16}$ | $76.18_{\pm2.62}$ | $62.05_{\pm2.16}$ | $60.31_{\pm4.95}$ | $52.35_{\pm11.02}$ | $48.61_{\pm2.34}$ | $55.10_{\pm10.57}$ |
| LRI-BERNOULLI | $74.38_{\pm4.33}$ | $81.42_{\pm1.52}$ | $78.23_{\pm1.11}$ | $65.64_{\pm2.44}$ | $89.22_{\pm3.58}$ | $68.76_{\pm7.35}$ | $54.87_{\pm1.89}$ | $72.12_{\pm2.60}$ |
| GMT-LIN | $\mathbf{77.45}_{\pm1.69}$ | $\mathbf{81.81}_{\pm1.57}$ | $\mathbf{79.17}_{\pm0.82}$ | $\mathbf{68.94}_{\pm1.08}$ | $\mathbf{96.17}_{\pm1.44}$ | $\mathbf{86.33}_{\pm6.16}$ | $59.70_{\pm1.10}$ | $70.62_{\pm3.59}$ |
| GMT-SAM | $75.61_{\pm1.86}$ | $81.96_{\pm1.35}$ | $78.28_{\pm1.34}$ | $65.69_{\pm2.61}$ | $93.93_{\pm3.59}$ | $83.20_{\pm4.74}$ | $\mathbf{60.03}_{\pm1.02}$ | $\mathbf{72.56}_{\pm2.27}$ |

image-converted datasets. Despite distribution shifts, `GMT-sam` still brings improvements up to $13\%$ with GIN, and up to $16\%$ against `GSAT` in Spurious-Motif.

**Results on geometric benchmarks.** Tables 3 and 4 shows the interpretation and generalization performance of various methods. Again, we also observe consistent non-trivial improvements of `GMT-lin` and `GMT-sam` in most cases than `GSAT` and post-hoc methods. Interestingly, `GMT-lin` leads to more improvements than `GMT-sam` in terms of interpretation performance despite of its simple modifications, while having a competitive generalization performance as `LRI`. In terms of generalization performance, `GMT-sam` remain the best method. The results on geometric datasets further demonstrate the strong generality of `GMT` across different tasks and backbones.

Table 4: Prediction performance (AUC) on the geometric learning datasets. The shadowed entries are the results with the mean-1*std larger than the mean of the best baselines.

| | ACTSTRACK | TAU3MU | SYNMOL | PLBIND | | ACTSTRACK | TAU3MU | SYNMOL | PLBIND |
|---|---|---|---|---|---|---|---|---|---|
| ERM | $97.40_{\pm0.32}$ | $82.75_{\pm0.16}$ | $99.30_{\pm0.20}$ | $85.31_{\pm2.21}$ | GMT-LIN | $93.92_{\pm0.98}$ | $82.60_{\pm0.17}$ | $99.26_{\pm0.27}$ | $86.29_{\pm0.80}$ |
| LRI-BERNOULLI | $94.00_{\pm0.78}$ | $86.36_{\pm0.06}$ | $99.30_{\pm0.15}$ | $85.80_{\pm0.70}$ | GMT-SAM | $\mathbf{98.55}_{\pm0.11}$ | $\mathbf{86.42}_{\pm0.08}$ | $\mathbf{99.89}_{\pm0.03}$ | $\mathbf{87.19}_{\pm1.86}$ |

(a) Counterfactual fidelity.     (b) Interpretation sensitivity.     (c) Generalization sensitivity.

Figure 3: Ablation studies.

**Ablation studies.** In complementary to the interpretability and generalizability study, we conduct further ablation studies to better understand the results. Fig. 3(a) shows the counterfactual fidelity of `GSAT`, `GMT-lin` and `GMT-sam` in Spurious-Motif (SPmotif) test sets. As shown in Fig. 3(a) that `GSAT` achieves a lower counterfactual fidelity. In contrast, `GMT-lin` and `GMT-sam` improve a higher counterfactual fidelity, which explains the reason for the improved interpretability of `GMT`. We also examine the hyperparameter sensitivity of `GMT-sam` in SPMotif-0.5 dataset. As shown in Fig. 3(b), 3(c), `GMT-sam` maintains strong robustness against the hyperparameter choices. The interpretation performance gets improved along with the sampling rounds, while a too larger GIB information regularizer weights will affect the optimization of `GMT` as well as the generalizability.

## 7 CONCLUSIONS

We developed a theoretical framework to analyze the expressive power of XGNNs by formulating the subgraph learning with multilinear extension (`SubMT`). We find that existing attention-based XGNNs will fail to approximate `SubMT`, which will lead to unfaithful interpretation as well as poor (OOD) generalization. To mitigate the issue, we propose a simple yet novel architecture called `GMT` which is provably more powerful in approximating `SubMT`. Extensive experiments on both graph classification and geometric learning benchmarks verify the superior interpretability and generalizability of `GMT`.

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

# Appendix of GMT

CONTENTS

# A  NOTATIONS

In the following, we list notations for key concepts that have appeared in this paper.

Table 5: Notations for key concepts involved in this paper.

| | |
|---|---|
| $\mathcal{G}$ | the graph space |
| $\mathcal{G}_c$ | the space of subgraphs with respect to the graphs from $\mathcal{G}$ |
| $\mathcal{Y}$ | the label space |
| $\rho$ | the pooling function of the GNN |
| $d(\cdot,\cdot)$ | a distribution distance metric |
| $L(\cdot,\cdot)$ | the loss function |
| $G \in \mathcal{G}$ | a graph |
| $G = (A, X)$ | a graph with the adjacency matrix $A \in \{0,1\}^{n \times n}$ and node feature matrix $X \in \mathbb{R}^{n \times d}$
for brevity, we also use $G$ and $Y$ to denote the random variables as the graphs and labels |
| $f = f_c \circ g$ | a XGNN with a subgraph extractor $g$ and a classifier $f_c$ |
| $g$ | a subgraph extractor $g : \mathcal{G} \to \mathcal{G}_c$ |
| $f_c$ | a classifier GNN $f_c : \mathcal{G}_c \to \mathcal{Y}$ |
| $G_c$ | the invariant subgraph with respect to $G$ |
| $G_s$ | the spurious subgraph with respect to $G$ |
| $\widehat{A}_c, \widehat{A}$ | the weighted adjacency matrix for causal subgraph with entries $A_{u,v} = \alpha_e$
as the sampling probability predicted by $g$ |
| $\widehat{A}_s$ | the weighted adjacency matrix for spurious subgraph with entries $A_{u,v} = 1 - \alpha_e$
as the sampling probability predicted by $g$ |
| $\widehat{G}_c$ | the estimated invariant subgraph produced by $g$
if the subgraph partitioning is conducted in an edge-centric view, then $\widehat{G}_c = (X, \widehat{A}_c)$ |
| $\widehat{G}_s$ | the estimated spurious subgraph produced by tacking the complementary of $\widehat{G}_c$
if the subgraph partitioning is conducted in an edge-centric view, then $\widehat{G}_s = (X, \widehat{A}_s)$ |
| $I(G_c; Y)$ | mutual information between the extracted subgraph $G_c$ and $Y$, specialized for maximizing $I(G;Y)$ |
| $P(G_c\|G) \in \mathbb{R}_+$ | the probability for sampling $G_c$ from $G$ with the subgraph extractor $g$ |
| $P(Y\|G) \in \mathbb{R}_+^{\|\mathcal{Y}\|}$ | the label distribution of $Y$ conditioned on $G$ |
| $P_f(Y\|G) \in \mathbb{R}_+^{\|\mathcal{Y}\|}$ | the predicted label distribution of $Y$ conditioned on $G$ |
| $f_c(G_c) \in \mathbb{R}_+^{\|\mathcal{Y}\|}$ | the predicted label distribution of $Y$ with $f_c$ by taking the input $G_c$ |

# B    MORE DETAILS ABOUT THE BACKGROUND

We begin by introducing related works in Appendix B.1 and then more backgrounds about graph information bottleneck in Appendix B.2, especially for how to obtain the formulas in the main text.

## B.1    MORE RELATED WORKS

We give a more detailed background introduction of interpretable and generalizable GNNs (XGNNs) in this section.

**Graph Neural Networks.**    We use $G = (A, X)$ to denote a graph with $n$ nodes and $m$ edges. Within $G$, $A \in \{0, 1\}^{n \times n}$ is the adjacency matrix, and $X \in \mathbb{R}^{n \times d}$ is the node feature matrix with a node feature dimension of $d$. This work focuses on the task of graph classification. Specifically, we are given a set of $N$ graphs $\{G_i\}_{i=1}^N \subseteq \mathcal{G}$ and their labels $\{Y_i\}_{i=1}^N \subseteq \mathcal{Y} = \mathbb{R}^c$ from $c$ classes. Then, we need to train a GNN $\rho \circ h$ with an encoder $h : \mathcal{G} \to \mathbb{R}^h$ that learns a meaningful representation $h_G$ for each graph $G$ to help predict their labels $y_G = \rho(h_G)$ with a downstream classifier $\rho : \mathbb{R}^h \to \mathcal{Y}$. The representation $h_G$ is typically obtained by performing pooling with a READOUT function on the learned node representations:

$$h_G = \text{READOUT}(\{h_u^{(K)} | u \in V\}), \tag{13}$$

where the READOUT is a permutation invariant function (e.g., SUM, MEAN) (Xu et al., 2019), and $h_u^{(K)}$ stands for the node representation of $u \in V$ at $K$-th layer that is obtained by neighbor aggregation:

$$h_u^{(K)} = \sigma(W_K \cdot a(\{h_v^{(K-1)}\} | v \in \mathcal{N}(u) \cup \{u\})), \tag{14}$$

where $\mathcal{N}(u)$ is the set of neighbors of node $u$, $\sigma(\cdot)$ is an activation function, e.g., ReLU, and $a(\cdot)$ is an aggregation function over neighbors, e.g., MEAN.

**Interpretable GNNs.**    Let $G = (A, X)$ be a graph with node set $V = \{v_1, v_2, ..., v_n\}$ and edge set $E = \{e_1, e_2, ..., e_m\}$, where $A \in \{0, 1\}^{n \times n}$ is the adjacency matrix and $X \in \mathbb{R}^{n \times d}$ is the node feature matrix. In this work, we focus on interpretable GNNs (or XGNNs) for the graph classification task, while the results can be generalized to node-level tasks as well (Wu et al., 2020). Given each sample from training data $\mathcal{D}_{\text{tr}} = (G^i, Y^i)$, an interpretable GNN $f := h \circ g$ aims to identify a (causal) subgraph $G_c \subseteq G$ via a subgraph extractor GNN $g : \mathcal{G} \to \mathcal{G}_c$, and then predicts the label via a subgraph classifier GNN $f_c : \mathcal{G}_c \to \mathcal{Y}$, where $\mathcal{G}, \mathcal{G}_c, \mathcal{Y}$ are the spaces of graphs, subgraphs, and the labels, respectively (Yu et al., 2021). Although *post-hoc* explanation approaches also aim to find an interpretable subgraph as the explanation for the model prediction (Ying et al., 2019; Yuan et al., 2020a; Vu & Thai, 2020; Luo et al., 2020; Yuan et al., 2021; Lin et al., 2021; 2022a), they are shown to be suboptimal in interpretation performance and sensitive to the performance of the pre-trained GNNs (Miao et al., 2022). Therefore, this work focuses on *intrinsic interpretable* GNNs (XGNNs).

A predominant approach to implement XGNNs is to incorporate the idea of information bottleneck (Tishby et al., 1999), such that $G_c$ keeps the minimal sufficient information of $G$ about $Y$ (Yu et al., 2021; 2022; Miao et al., 2022; 2023; Yang et al., 2023), which can be formulated as

$$\max_{G_c} I(G_c; Y) - \lambda I(G_c; G), \ G_c \sim g(G), \tag{15}$$

where maximizing the mutual information between $G_c$ and $Y$ endows the interpretability of $G_c$ while minimizing $I(G_c; G)$ ensures $G_c$ captures only the most necessary information, $\lambda$ is a hyperparamter trade off between the two objectives. In addition to minimizing $I(G_c; G)$, there are also alternative approaches that impose different constraints such as causal invariance (Chen et al., 2022a; Li et al., 2022) or disentanglement (Wu et al., 2022b; Sui et al., 2022; Liu et al., 2022a; Fan et al., 2022) to identify the desired subgraphs. When extracting the subgraph, XGNNs adopts the attention mechanism to learn the sampling probability of each edge or node, which avoids the complicated Monte Carlo tree search used in other alternative implementations (Zhang et al., 2022). Specifically, given node representation learned by message passing $H_i \in \mathbb{R}^h$ for each node $i$, XGNNs either learns a **node attention** $\alpha_i \in \mathbb{R}_+ = \sigma(a(H_i))$ via the attention function $a : \mathbb{R}^h \to \mathbb{R}_+$, or the **edge attention** $\alpha_e \in \mathbb{R}_+ = \sigma(a([H_u, H_v]))$ for each edge $e = (u, v)$ via the attention function $a : \mathbb{R}^{2h} \to \mathbb{R}_+$, where $\sigma(\cdot)$ is a sigmoid function. $\boldsymbol{\alpha} = [\alpha_1, ..., \alpha_m]^T$ essentially elicits a subgraph

distribution of the interpretable subgraph. In this work, we focus on edge attention-based subgraph distribution as it is most widely used in XGNNs while our method can be easily generalized to node attention-based subgraph approaches as demonstrated in the experiments with geometric learning datasets.

Besides, Fountoulakis et al. (2023); Lee et al. (2023) find the failures of graph attention networks in properly propagating messages with the attention mechanism. They differ from our work as they focus on node classification tasks.

**Faithful interpretation and (OOD) generalization.** The faithfulness of interpretation is critical to all interpretable and explainable methods (Ribeiro et al., 2016; Lipton, 2018; Alvarez-Melis & Jaakkola, 2018; Rudin, 2018; Jain & Wallace, 2019; Karimi et al., 2023). Yet, there are many failure cases found especially when with attention mechanisms. For example, Jain & Wallace (2019) reveals that in NLP, randomly shuffling or imposing adversarial noises will not affect the predictions too much, highlighting a weak correlation between attention and prediction. Karimi et al. (2023) present a causal analysis showing the hyperparameters and the architecture setup could be a cofounder that affects the causal analysis. Chang et al. (2020) show interpretations will fail when distribution shifts are presented. Although the faithfulness of explanation/interpretations has been widely a concern for Euclidean data, whether and how GNNs and XGNNs suffer from the same issue is under-explored.

Talking about the progress in graph data, there are several metrics developed to measure the faithfulness of graph explanations, such as fidelity (Yuan et al., 2020b; Amara et al., 2022), counterfactual robustness (Bajaj et al., 2021; Prado-Romero et al., 2022; Ma et al., 2022), and equivalence (Crabbé & van der Schaar, 2023), which are however limited to post-hoc graph explanation methods. In fact, post-hoc explanation methods are mostly developed to adhere the faithfulness measures such as fidelity. However, as shown by Miao et al. (2022), the post-hoc methods are suboptimal in finding the interpretable subgraph and sensitive to the pre-trained model, which highlights a drawback of the existing faithfulness measure. In contrast, we develop the first faithfulness measure for XGNNs in terms of counterfactual invariance.

Although Bajaj et al. (2021); Prado-Romero et al. (2022); Ma et al. (2022) also adopt the concept of counterfactual to develop post-hoc explanation methods, they focus on finding the minimal perturbations that will change the predictions. Counterfactual is also widely used to improve graph representation learning (Guo et al., 2023). In contrast, we adopt the concept of counterfactual to measure the sensitivity of the XGNNs predictions to the predicted attention.

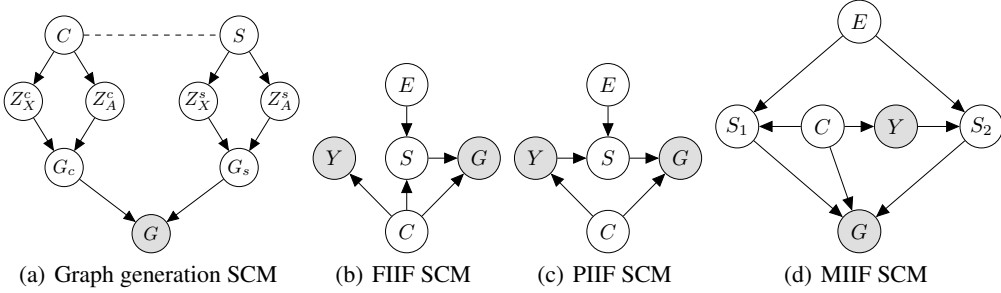

(a) Graph generation SCM   (b) FIIF SCM   (c) PIIF SCM   (d) MIIF SCM

Figure 4: Full SCMs on Graph Distribution Shifts (Chen et al., 2022a).

**On the natural connection of XGNNs and OOD generalization on graphs.** In the context of graph classification, the generalization ability and the faithfulness of the interpretation are naturally intertwined in XGNNs. In many realistic graph classification practices such as drug property prediction (Ji et al., 2022; Zhang et al., 2023), the property of a drug molecule can naturally be represented by a subgraph, termed as causal subgraph. The causal subgraph, in return, holds a causal relationship with the drug property. Therefore, it is natural to identify the underlying causal subgraph to provide OOD generalizable predictions and interpretations.

Typically, XGNNs need to extract the underlying ground truth subgraph in order to make correct predictions on unseen test graphs (Miao et al., 2022). When distribution shifts are presented in the

test data, it is critical to find the underlying subgraph that has a causal relationship with the target label (or causal subgraphs) (Chang et al., 2020; Chen et al., 2022a).

We now briefly introduce the background of causal subgraph and OOD generalization. Specifically, we are given a set of graph datasets $\mathcal{D} = \{\mathcal{D}_e\}_e$ collected from multiple environments $\mathcal{E}_{\text{all}}$. Samples $(G_i^e, Y_i^e) \in \mathcal{D}^e$ from the same environment are considered as drawn independently from an identical distribution $\mathbb{P}^e$. We consider the graph generation process proposed by Chen et al. (2022a) that covers a broad case of graph distribution shifts. Fig. 4 shows the full graph generation process considered in Chen et al. (2022a). The generation of the observed graph $G$ and labels $Y$ are controlled by a set of latent causal variable $C$ and spurious variable $S$, i.e.,

$$G := f_{\text{gen}}(C, S).$$

$C$ and $S$ control the generation of $G$ by controlling the underlying invariant subgraph $G_c$ and spurious subgraph $G_s$, respectively. Since $S$ can be affected by the environment $E$, the correlation between $Y, S$ and $G_s$ can change arbitrarily when the environment changes. $C$ and $S$ control the generation of the underlying invariant subgraph $G_c$ and spurious subgraph $G_s$, respectively. Since $S$ can be affected by the environment $E$, the correlation between $Y, S$ and $G_s$ can change arbitrarily when the environment changes. Besides, the latent interaction among $C, S$ and $Y$ can be further categorized into *Full Informative Invariant Features* (*FIIF*) when $Y \perp\!\!\!\perp S|C$ and *Partially Informative Invariant Features* (*PIIF*) when $Y \not\perp\!\!\!\perp S|C$. Furthermore, PIIF and FIIF shifts can be mixed together and yield *Mixed Informative Invariant Features* (*MIIF*), as shown in Fig. 4. We refer interested readers to Chen et al. (2022a) for a detailed introduction to the graph generation process.

To tackle the OOD generalization challenge on graphs generated following in Fig. 4, the existing invariant graph learning approaches generically aim to identify the underlying invariant subgraph $G_c$ to predict the label $Y$ (Wu et al., 2022a; Chen et al., 2022a). Specifically, the goal of OOD generalization on graphs is to learn an invariant XGNN $f := f_c \circ g$, with the following objective:

$$\max_{f_c, g} I(\widehat{G}_c; Y), \text{ s.t. } \widehat{G}_c \perp\!\!\!\perp E, \ \widehat{G}_c = g(G). \tag{16}$$

Since $E$ is not observed, many strategies are proposed to impose the independence of $\widehat{G}_c$ and $E$. A common approach is to augment the environment information. For example, based on the estimated invariant subgraphs $\widehat{G}_c$ and spurious subgraphs $\widehat{G}_s$, Wu et al. (2022b); Liu et al. (2022a); Wu et al. (2022a); Yu et al. (2023) propose to generate new environments, while Li et al. (2022) propose to infer the underlying environment labels via clustering. Yang et al. (2022) propose a variational framework to infer the environment labels. Gui et al. (2023) propose to learn causal independence between labels and environments. Yu et al. (2021; 2022); Miao et al. (2022; 2023); Yang et al. (2023) adopt graph information bottleneck to tackle FIIF graph shifts, and they cannot generalize to PIIF shifts. Nevertheless, since most of the existing works adopt the backbone of XGNNs, and XGNNs with information bottleneck is the state-of-the-art method with both high interpretation performance and OOD generalization performance, the focus in this work will be around tackling FIIF shifts with the principle of graph information bottleneck. More details are given in the next section.

In addition to the aforementioned approaches, Yehudai et al. (2021); Bevilacqua et al. (2021); Zhou et al. (2022) study the OOD generalization as an extrapolation from small graphs to larger graphs in the task of graph classification and link prediction. In contrast, we study OOD generalization against various graph distribution shifts formulated in Fig. 4. Li et al. (2023) propose an extrapolation strategy to improve OOD generalization on graphs. In addition to the standard OOD generalization tasks studied in this paper, Xu et al. (2021); Mahdavi et al. (2022) study the OOD generalization in tasks of algorithmic reasoning on graphs. Jin et al. (2022) study the test-time adaption in the graph regime. Kamhoua et al. (2022) study the 3D shape matching under the presence of noises.

**Multilinear extension.** Multilinear extension serves as a powerful tool for maximizing combinatorial functions, especially for submodular set function maximization (Owen, 1972; Călinescu et al., 2007; Vondrak, 2008; Calinescu et al., 2011; Chekuri et al., 2014; 2015; Bian et al., 2019; Sahin et al., 2020; Bian et al., 2022; Karalias et al., 2022). For example, Vondrak (2008); Calinescu et al. (2011) study the multilinear extension in the context of social welfare. Bian et al. (2022) study the multilinear extension for cooperative games. It is the expected value of a set function under the fully factorized i.i.d. Bernoulli distribution. The closest work to ours is Karalias et al. (2022) that builds neural set function extensions for multiple discrete functions. Nevertheless, to the best of our knowledge, the notion of multilinear extensions for XGNNs is yet underexplored. In contrast, in this work, we

are the first to identify subgraph multilinear extension as the factorized subgraph distribution for interpretable subgraph learning.

## B.2 VARIATIONAL BOUNDS AND REALIZATION OF THE IB PRINCIPLE

We first introduce how to derive Eq. 3 in the main text, and then discuss how to implement the graph information bottleneck regularization $\min I(G_c; G)$ following the state-of-the-art architecture GSAT (Miao et al., 2022; 2023).

**Variational bounds for $I(G; Y)$.** For the term $I(G; Y)$, notice that

$$I(G; Y) = \mathbb{E}_{G,Y}\left[\log \frac{P(Y|G)}{P(Y)}\right] \tag{17}$$

Since the true $P(Y|G)$ is intractable, through XGNN modelling we introduce a variational approximation $P_{f_c,g}(Y|G)$. Then,

$$I(G; Y) = \mathbb{E}_{G,Y}\left[\log \frac{P_{f_c,g}(Y|G)}{P(Y)}\right] + \mathbb{E}_{G,Y}\left[\log \frac{P(Y|G)}{P_{f_c,g}(Y|G)}\right] \tag{18}$$

$$= \mathbb{E}_{G,Y}\left[\log \frac{P_{f_c,g}(Y|G)}{P(Y)}\right] + D_{\mathrm{KL}}(P(Y|G)||P_{f_c,g}(Y|G)) \tag{19}$$

$$\geq \mathbb{E}_{G,Y}\left[\log P_{f_c,g}(Y|G)\right] + H(Y) \tag{20}$$

Since the optimization does not involve $H(Y)$, we continue with $\mathbb{E}_{G,Y}\left[\log P_{f_c,g}(Y|G)\right]$,

$$\mathbb{E}_{G,Y}\left[\log P_{f_c,g}(Y|G)\right] = \mathbb{E}_{G,Y}\left[\log \sum_{G_c} P_{f_c,g}(Y, G_c|G)\right] \tag{21}$$

$$= \mathbb{E}_{G,Y}\left[\log \sum_{G_c} P_{f_c,g}(Y|G, G_c)P_{f_c,g}(G_c|G)\right] \tag{22}$$

$$= \mathbb{E}_{G,Y}\left[\log \sum_{G_c} P_{f_c}(Y|G_c)P_g(G_c|G)\right] \tag{23}$$

where Eq. 23 is due to the implementation of XGNNs. Eq. 23 can also be written with expectations:

$$\mathbb{E}_{G,Y}\left[\log \sum_{G_c} P_{f_c}(Y|G_c)P_g(G_c|G)\right] = \mathbb{E}_{G,Y}\left[\log \mathbb{E}_{G_c \sim \mathbb{P}(G_c|G)} P_{f_c}(Y|G_c)\right].$$

Maximizing $I(G; Y)$ is then equivalent to minimizing $-I(G; Y)$, and further minimizing $\mathbb{E}_{G,Y}[-\log P_{f_c,g}(Y|G)]$. This achieves to Eq. 3 in the main text, i.e.,

$$\mathbb{E}_{(G,Y)\sim\mathcal{D}_{\mathrm{tr}}}[-\log P(Y|\mathbb{E}_{G_c \overset{g}{\sim} G} G_c)] = \mathbb{E}_{(G,Y)\sim\mathcal{D}_{\mathrm{tr}}}[L(f_c(\boldsymbol{\alpha}; G), Y)],$$

with $L$ as the cross entropy loss, and $\boldsymbol{\alpha}$ as the predicted sampling probability for edges. $\boldsymbol{\alpha}$ factorizes the sampling probability of the subgraphs as independent Bernoulli distributions on edges $e \sim \mathrm{Bern}(\alpha_e), \forall e \in E$:

$$P(G_c|G) = \prod_{e \in G_c} \alpha_e \prod_{e \in G/G_c} (1 - \alpha_e).$$

**Variational bounds for $I(G_c; G)$.** For the term $I(G_c; G)$, since we factorize graph distribution as multiple independent Bernoulli distributions on edges, we are able to calculate the KL divergence to upper bound $I(G_c; G)$:

$$I(G_c; G) \leq D_{\mathrm{KL}}(P(G_c|G)||Q(G_c)), \tag{24}$$

where $Q(G_c)$ is a variational approximation to $P(G_c)$. $D_{\mathrm{KL}}$ can be obtained via

$$D_{\mathrm{KL}}(P(G_c|G)||Q(G_c)) = \sum_{e \in G_c} D_{\mathrm{KL}}(\mathrm{Bern}(\alpha_e)||\mathrm{Bern}(r)) + c(n, r), \tag{25}$$

where $c(n, r)$ is a small constant, $r$ is a hyperparameter to specify the prior for subgraph distributions. To minimize $I(G_c; G)$ is essentially to minimize $D_{\text{KL}}(\text{Bern}(\alpha_e)||\text{Bern}(r))$. The KL divergence can be directly calculated as

$$D_{\text{KL}}(\text{Bern}(\alpha_e)||\text{Bern}(r)) = \sum_e \alpha_e \log \frac{\alpha_e}{r} + (1 - \alpha_e) \log \frac{(1 - \alpha_e)}{(1 - r)}. \tag{26}$$

Miao et al. (2022) find the mutual information based regularization can effectively regularize the information contained in $G_c$ than previous implementations such as vanilla size constraints with the norm of attention scores or connectivity constraints (Yu et al., 2021).

Besides, we would like to note that GSAT implementation provided by the author does not exactly equal to the mathematical formulation, i.e., they directly take the unormalized attention to Eq. 26, as acknowledged by the authors [3]. The reason for using another form of information regularization is because the latter empirically performs better. Nevertheless, LRI adopts the mathematically correct form and obtains better empirical performance. In our experiments, we adopt the mathematically correct form for both regular and geometric learning tasks, in order to align with the theory. Empirically, we find the two forms perform competitively well with the suggested hyparaemters and hence stick to the mathematically correct form.

## C  ON THE GENERALIZATION AND INTERPRETABILITY: A CAUSAL VIEW

### C.1  STRUCTURAL CAUSAL MODEL FOR XGNNS

We provide a detailed description and the full structural causal model of XGNNs in complementary to the causal analysis in Sec. 4.

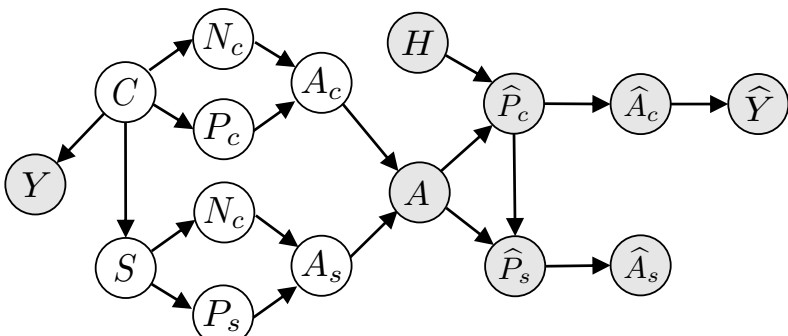

Figure 5: Bernoulli Parameterized SCM for interpretable GNN

**Data generation.**  We consider the same data model as previous works (Bevilacqua et al., 2021; Miao et al., 2022; Chen et al., 2022a), where the underlying causal subgraph $G_c$ and the spurious subgraph $G_s$ will be assembled via some underlying assembling process $G = f_g(G_c, G_s)$, as illustrated in Appendix B Fig. 4.

We focus on the FIIF distribution shifts (Fig. 4(b)) that can be resolved by graph information bottleneck (Miao et al., 2022; Chen et al., 2022a). As shown in the figure, there are latent causal and spurious variables $C$ and $S$ that have an invariant and spurious correlation with the label $Y$, respectively. $C$ and $S$ further control the generation of the graph structure of the causal subgraph $G_c$, and the spurious subgraph $G_s$. Specifically, $C$ and $S$ will specify the number of nodes in $G_c$ and $G_s$ as $N_c$ and $N_s$. Then, $C$ and $S$ further control the underlying Bernoulli distributions on edges, by specifying the sampling probability as $P_c$ and $P_s$. With $N_c$ and $P_c$ (or $N_s$ and $P_s$), $A_c$ (or $A_s$) can be sampled and then assembled into the observed graph structure $A$. As we focus on the edge-centric view, our discussion focuses on the graph structures $A_c$ and $A_s$ of the subgraphs. Nevertheless, a similar generation model can also be developed for the node-centric view.

---

[3] https://github.com/Graph-COM/GSAT/issues/10

**Interpretation.**   Correspondingly, XGNNs first uses a subgraph extractor to predict the causal and spurious subgraphs $\widehat{G}_c$ and $\widehat{G}_s$, respectively. The extraction aims to reverse the generation and recover the underlying $P_c$, by learning the $\widehat{P}_c$ via the attention $\boldsymbol{\alpha}$. We denote the architecture and the hyperparameter settings as $H$. Once $\widehat{P}_c$ is determined, $\widehat{P}_s = 1 - \widehat{P}_c$ is also obtained by finding the complementary part. Then, the estimated causal and spurious subgraphs are sampled from $\widehat{P}_c$ and $\widehat{P}_s$, respectively. With the estimated causal subgraph $\widehat{G}_c = (X, \widehat{A}_c)$, the classifier GNN $c(\cdot)$ will use it to make a prediction $\widehat{Y}$.

### C.2   PRACTICAL ESTIMATION OF COUNTERFACTUAL FIDELITY

Since it is prohibitively expensive to enumerate all possible $\widetilde{G}$ and the distance $\delta$ to examine the counterfactual fidelity. We instead consider an alternative notion that adopts random perturbation onto the learned attention score. Specifically, we consider a random attention matrix $\widetilde{A} \sim \sigma(\mathcal{N}(\mu_{\widehat{H}_A}, \sigma_{\widehat{H}_A}))$, where $\mu_{\widehat{H}_A}$ and $\sigma_{\widehat{H}_A}$ are the mean and standard deviation of the pre-attention matrix $\widehat{H}_A$ (The adjacency matrix with the unnormalized attention). Since each non-symmetric entry in the attention is generated independently, each non-symmetric entry in $\widetilde{A}$ is sampled independently following the factorization of $P(G)$. We randomly sample $\widetilde{A}$ by $k$ times and calculate the following:

$$c_{\widehat{G}_c} = \frac{1}{k} \sum_{i=1}^{k} d(f_c(Y|\widetilde{G}_c^i), f_c(Y|\widehat{G}_c)), \tag{27}$$

where $\widetilde{G}_c^i = (X, \widetilde{A}_c^i)$ and $d$ is total variation distance. The detailed computation of the practical counterfactual fidelity is provided in Algorithm 1.

---

**Algorithm 1** Practical estimation of counterfactual fidelity.

---

1: **Input:** Training data $\mathcal{D}_{\mathrm{tr}}$; a trained XGNN $f$ with subgraph extractor $g$, and classifier $f_c$; sampling times $e_s$; batch size $b$; total variation distance $d(\cdot)$;
2: // Minibatch sampling.
3: **for** $j = 1$ to $|\mathcal{D}_{\mathrm{tr}}|/b$ **do**
4:     Sample a batch of data $\{G^i, Y^i\}_{i=1}^{b}$ from $\mathcal{D}_{\mathrm{tr}}$;
5:     Obtain the pre-attention matrix $\widehat{H}_A$;
6:     Obtain the attention matrix $\widehat{A} = \sigma(\widehat{H}_A)$;
7:     Obtain the original prediction with $f_c$ based on the attention matrix $\widehat{A}$ as $\{\hat{y}^i\}_{i=1}^{b}$;
8:     // Random noises injection.
9:     **for** $k = 1$ to $e_s$ **do**
10:         Sample a random attention matrix $\widetilde{A} \sim \sigma(\mathcal{N}(\mu_{\widehat{H}_A}, \sigma_{\widehat{H}_A}))$;
11:         Obtain sampling attention $\{\boldsymbol{\alpha}^i\}_{i=1}^{b}$;
12:         Obtain the perturbed prediction with $f_c$ based on the attention matrix $\widetilde{A}$ as $\{\hat{y}_k^i\}_{i=1}^{b}$;
13:     **end for**
14:     Calculate $\{c_{\widehat{G}_c}^i\}_{i=1}^{b}$ with $k$ groups of $\{\hat{y}_k^i\}_{i=1}^{b}$ and $\{\hat{y}^i\}_{i=1}^{b}$;
15:     Obtain the averaged $c_{\widehat{G}_c}^j$ within the batch;
16: **end for**
17: Obtain the averaged $c_{\widehat{G}_c}$ within the training data;
18: **Return** estimated $c_{\widehat{G}_c}$;

---

Shown as in Fig. 6, 7, we plot the counterfactual fidelity of GSAT and the simulated SubMT with 10 and 100 sampling rounds on BA-2Motifs and Mutag datasets. The SubMT is approximated via GMT-sam with different sampling rounds. It can be found that GSAT achieves a counterfactual fidelity that is 2 to 3 times lower than the simulated SubMT via GMT-sam with 10 and 100 sampling rounds. Moreover, in simple tasks such as BA-2Motifs and Mutag, using larger sampling rounds like 100 does not necessarily bring more counterfactual fidelity. One reason can be using small sampling rounds to touch the upper bounds of counterfactual fidelity measured in our work. We also provide a discussion on why the counterfactual fidelity grows slowly at the initial epochs in BA-2Motif datasets in Appendix E.2. More counterfactual fidelty studies can be found in Appendix F.5.

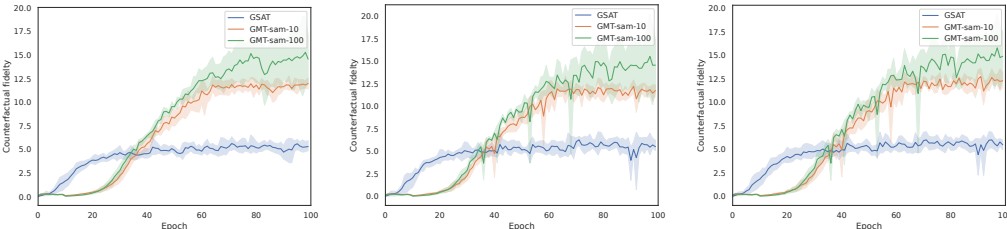

(a) `SubMT` on BA-2Motifs trainset. (b) `SubMT` on BA-2Motifs valset. (c) `SubMT` on BA-2Motifs test set.

Figure 6: Counterfactual fidelity on BA-2Motifs.

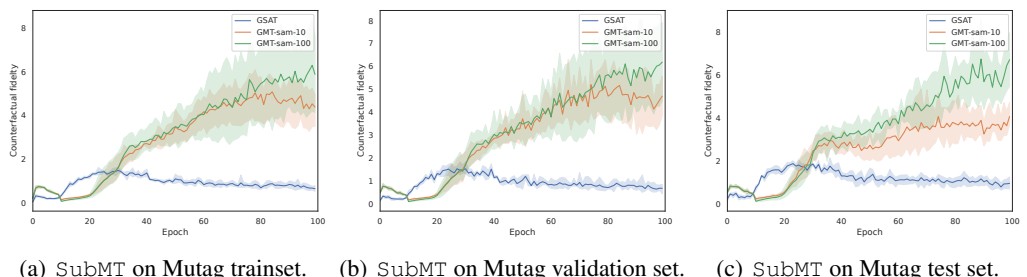

(a) `SubMT` on Mutag trainset. (b) `SubMT` on Mutag validation set. (c) `SubMT` on Mutag test set.

Figure 7: Counterfactual fidelity on Mutag.

## D THEORIES AND PROOFS

### D.1 USEFUL DEFINITIONS

We give the relevant definitions here for ease of reference when reading our proofs.

**Definition D.1** (Subgraph multilinear extension (`SubMT`)). *Given the attention $\boldsymbol{\alpha} \in \mathbb{R}_+^m$ as edge sampling probability of $G_c$, XGNNs factorize $P(G)$ as independent Bernoulli distributions on edges:*

$$P(G_c|G) = \prod_{e \in G_c} \alpha_e \prod_{e \in G/G_c} (1 - \alpha_e),$$

*which elicits the multilinear extension of $f_c(G_c)$ in Eq. 3 as:*

$$F_c(\boldsymbol{\alpha}; G) := \sum_{G_c \in G} f_c(G_c) \prod_{e \in G_c} \alpha_e \prod_{e \in G/G_c} (1 - \alpha_e) = \mathbb{E}_{G_c \overset{g}{\sim} G} f_c(G_c). \tag{28}$$

**Definition D.2** ($\epsilon$-`SubMT` approximation). *Let $d(\cdot, \cdot)$ be a distribution distance metric, a XGNN $f = f_c \circ g$ $\epsilon$-approximates SubMT (Def. 3.1), if there exists $\epsilon \in \mathbb{R}_+$ such that $d(P_f(Y|G), P(Y|G)) \leq \epsilon$ where $P(Y|G) \in \mathbb{R}^{|\mathcal{Y}|}$ is the ground truth conditional label distribution, and $P_f(Y|G) \in \mathbb{R}^{|\mathcal{Y}|}$ is the predicted label distribution for $G$ via a XGNN $f$, i.e., $P_f(Y|G) = f_c(\mathbb{E}_{G_c \overset{g}{\sim} G} G_c)$.*

**Definition D.3** (($\delta, \epsilon$)-counterfactual fidelity). *Given a meaningful minimal distance $\delta > 0$, let $d(\cdot, \cdot)$ be a distribution distance metric , if a XGNN $f = f_c \circ g$ commits to the $\epsilon$−counterfactual fidelity, then there exist $\epsilon > 0$ such that, $\forall G, \widetilde{G}$ that $d(P(Y|G), P(Y|\widetilde{G})) \geq \delta$, the following holds:*

$$d(P_f(Y|\widetilde{G}), P_f(Y|G)) \geq \epsilon\delta.$$

### D.2 PROOF FOR PROPOSITION 3.3

Obviously $f_c(\cdot)$ is a non-linear function even with a linearized GNN (Wu et al., 2019) with linear activations and pooling such as sum pooling, which can be written as:

where $\rho$ is the pooling, $k$ is the number of layers and $\boldsymbol{W} \in \mathbb{R}^{h \times h}$ are the learnable weights.

**Proposition D.4.** *Consider a linearized GNN (Wu et al., 2019) with number of message passing layers $k > 1$, linear activations and pooling,*

$$f_c(G_c) = \rho(\widehat{A}^k X \boldsymbol{W}), \tag{29}$$

*if there exists $1 \le i, j \le n$ that $0 < \widehat{A}_{i,j} < 1$, Eq. 8 can not hold, thus Eq. 29 can not approximate* SubMT *(Def. 3.1).*

*Proof.* To begin with, given a linear pooling function $\rho$, one could write the outcomes of $f_c(A) = \rho(A^k X W)$ as a summation in $A_{i,j}^k v_{i,j}$, with $v_{i,j}$ is the weight that accounting for the pooling as well as $XW$:

$$f_c(A) = \sum_i \sum j A_{i,j} v_{i,j}. \tag{30}$$

Given the linearity of expectations, the comparison between $E[f_c(A)]$ and $f_c(E[A])$ now turns into the comparison between $E[A_{i,j}^k v_j]$ and $(E[A_{i,j}])^k v_j$. Since $A_i j$ is drawn from the Bernoulli distribution, with the expectation as $\widehat{A}_{i,j}$, it suffices to know that

$$E[A_{i,j}^k v_j] = 1^k \widehat{A}_{i,j} + 0^k (1 - \widehat{A}_{i,j}) = \widehat{A}_{i,j}, \tag{31}$$

while $(E[A_{i,j}])^k = \widehat{A}_{i,j}^k$. Then, we know that $E[f_c(A)] \ne f_c(E[A])$. $\qquad \square$

We also conduct empirical verifications with GSAT implemented in GIN and SGC with various layers in Appendix F.6.

### D.3    PROOF FOR PROPOSITION 4.2

**Proposition D.5.** *If a XGNN $f$ $\epsilon$-approximates* SubMT *(Def. D.2), then $f$ also satisfies $(\delta, 1 - \frac{2\epsilon}{\delta})$-counterfactual fidelity (Def. D.3).*

*Proof.* Considering any two graphs $G$ and $\widetilde{G}$ that $d(P(Y|G), P(Y|\widetilde{G}) \ge \delta$, since $d$ is a distance metric, we have the following inequality holds:

$$d(P(Y|G), P_f(Y|\widetilde{G})) \le d(P_f(Y|G), P(Y|G)) + d(P_f(Y|G), P_f(Y|\widetilde{G})), \tag{32}$$

by the triangle inequality. Furthermore, we have

$$d(P(Y|G), P_f(Y|\widetilde{G})) - d(P_f(Y|G), P(Y|G)) \le d(P_f(Y|G), P_f(Y|\widetilde{G})) \tag{33}$$

As XGNN $f$ that $\epsilon$-approximates SubMT, we have the following by definition:

$$d(P_f(Y|\widetilde{G}), P(Y|\widetilde{G})) \le \epsilon, d(P_f(Y|G), P(Y|G)) \le \epsilon.$$

Then, call the triangle inequality again, we have

$$
\begin{aligned}
d(P(Y|G), P(Y|\widetilde{G})) &\le d(P_f(Y|\widetilde{G}), P(Y|G)) + d(P_f(Y|\widetilde{G}), P(Y|\widetilde{G})) \\
d(P(Y|G), P(Y|\widetilde{G})) - d(P_f(Y|\widetilde{G}), P(Y|\widetilde{G})) &\le d(P_f(Y|\widetilde{G}), P(Y|G)) \\
\delta - d(P_f(Y|\widetilde{G}), P(Y|\widetilde{G})) &\le d(P_f(Y|\widetilde{G}), P(Y|G)) \\
\delta - \epsilon &\le d(P_f(Y|\widetilde{G}), P(Y|G)).
\end{aligned}
\tag{34}
$$

Combining the aforementioned three inequalities, we have

$$d(P_f(Y|\widetilde{G}), P(Y|G)) - d(P_f(Y|G), P(Y|G)) \ge \delta - 2\epsilon,$$

Then, it suffices to know that

$$\delta - 2\epsilon \le d(P_f(Y|G), P_f(Y|\widetilde{G})). \tag{35}$$

$\qquad \square$

### D.4 PROOF FOR THEOREM 5.1

**Theorem D.6.** *Given the attention matrix $\widehat{A}$, and the distribution distance metric $d$ as the total variation distance, let $C = |\mathcal{Y}|$, for a* `GMT-sam` *with $t$ i.i.d. samples of $G_c^i \sim P(G_c|G)$, then, there exists $\epsilon \in \mathbb{R}_+$ such that, with a probability at least $1 - e^{-t\epsilon^2/4}$,* `GMT-sam` *$\frac{\epsilon C}{2}$-approximates* `SubMT` *(Def. D.2) and satisfies $(\delta, 1 - \frac{\epsilon C}{\delta})$ counterfactual fidelity (Def. D.3).*

*Proof.* Recall the `SubMT` objective:

$$F_c(\boldsymbol{\alpha}; G) := \sum_{G_c \in G} f_c(G_c) \prod_{e \in G_c} \alpha_e \prod_{e \in G/G_c} (1 - \alpha_e),$$

which is the expanded form of $\mathbb{E}[f_c(G_c)]$, $G_c \sim P(G_c|\widehat{A})$. Then, denote $M = \max|f_c(G_c)|$, $f_c(G_c)$ can be considered as a random variable within the range of $[-M, M]$. Considering $t$ random i.i.d. examples of $\{G_c^i\}_{i=1}^t$ drawn from $P(G_c|\widehat{A})$, and the predicted probability for each class, denoted as $Y_i = \frac{1}{M} f_c(G_c^i)$, we then have $Y_i \in [-1, 1]$ and $\sum_{i=1}^t \mathbb{E}[Y_i] = \frac{t}{M} F(\boldsymbol{\alpha}; G)$. It allows us to adopt the Markov's inequality and obtain the following Chernoff bound:

$$\Pr(|\sum_{i=1}^t Y_i - \frac{t}{M} F(\boldsymbol{\alpha}; G) > t\epsilon|) < e^{-t^2\epsilon^2/4t} = e^{-t\epsilon^2/4}.$$

Since by definition of `GMT-sam`, i.e.,

$$f_c^s(\widehat{G}_c) = \frac{1}{t} \sum_{i=1}^t f_c(Y|G_c^i),$$

we have

$$\sum_{i=1}^t Y_i = \frac{t}{M} \sum_{i=1}^t f_c(G_c^i) = \frac{t}{M} f_c^s(\widehat{G}_c),$$

the bound can be written as:

$$\Pr(|\frac{t}{M} f_c^s(\widehat{G}_c) - \frac{t}{M} F(\boldsymbol{\alpha}; G) > t\epsilon|) < e^{-t^2\epsilon^2/4t} = e^{-t\epsilon^2/4}$$
$$\Pr(|f_c^s(\widehat{G}_c) - F(\boldsymbol{\alpha}; G) > \epsilon M|) < e^{-t\epsilon^2/4} \tag{36}$$
$$\Pr(|f_c^s(\widehat{G}_c) - F(\boldsymbol{\alpha}; G) \le \epsilon M|) \ge 1 - e^{-t\epsilon^2/4}.$$

In other words, with a probability at least $1 - e^{-t\epsilon^2/4}$, we have the following holds:

$$|f_c^s(\widehat{G}_c) - F_c(\boldsymbol{\alpha}; G)]| \le \epsilon M. \tag{37}$$

Since $M$ is defined as the maximal probability for each class,

$$M = \max \mathbb{E}[f_c P(Y|G_c)],$$

it suffices to know that $M \le 1$. Therefore, it follows that

$$|f_c^s(\widehat{G}_c) - F_c(\boldsymbol{\alpha}; G)]| \le \epsilon,$$

for each class, which further implies that

$$|f_c^s(\widehat{G}_c) - F_c(\boldsymbol{\alpha}; G)]| \le \epsilon|\mathcal{Y}| = \epsilon C,$$

which commits to the $\frac{\epsilon C}{2}$ `SubMT` approximation under the total variation distance. Then, using the results of Proposition 4.2, we know `GMT-sam` also commits to the $1 - \frac{\epsilon C}{\delta}$ counterfactual fidelity. $\quad\square$

# E   MORE DISCUSSIONS ON PRACTICAL IMPLEMENTATIONS OF GMT

We provide more discussion in complementary to the description of Sec. 5 in the main text.

## E.1   ALGORITHMS OF GMT

**Training subgraph extractor with random subgraph sampling.**   We focus on discussing the implementation details of GMT-sam since GMT-lin differs from GSAT only in the number of weighted message passing times. GMT-sam contains two stages: i) subgraph extractor training, and ii) neural subgraph extension learning. The first stage aims to train the subgraph extractor to extract the desired subgraphs, while the second stage aims to reduce the additional computation overhead of the random subgraph sampling, and further better learn the correlations between the soft subgraphs and the labels. The algorithm for stage i) is given in Algorithm 2 and for stage ii) is given in Algorithm 3, respectively.

---

**Algorithm 2** Subgraph extractor training algorithm of **G**raph **M**ultilinear ne**T** (**GMT**).

1: **Input:** Training data $\mathcal{D}_{\text{tr}}$; a XGNN $f$ with subgraph extractor $g$, and classifier $f_c$; subgraph sampling epochs $e_s$; length of maximum subgraph learning epochs $e_l$; batch size $b$; loss function $l(\cdot)$; subgraph regularization $o(\cdot)$; subgraph regularization weight $\gamma$;
2: Randomly initialize $f$;
3: // Phase of subgraph learning.
4: **for** $j = 1$ to $e_l$ **do**
5:     Sample a batch of data $\{G^i, Y^i\}_{i=1}^b$ from $\mathcal{D}_{\text{tr}}$;
6:     Obtain sampling attention $\{\boldsymbol{\alpha}^i\}_{i=1}^b$ via Eq. 38;
7:     // MCMC subgraph sampling.
8:     **for** $k = 1$ to $e_s$ **do**
9:         Obtain the sampling probability $\{\boldsymbol{\beta}^i\}_{i=1}^b$ via Eq. 39 using Gumbel-softmax;
10:         Randomly sample subgraphs $\{G_c^i \sim \text{Ber}(\boldsymbol{\beta}^i)\}_{i=1}^b$ via Eq. 40;
11:         Obtain predictions as logits $\{\hat{y}_j^i\}_{i=1}^b$;
12:     **end for**
13:     Obtain simulated prediction $\{\hat{y}^i = \frac{1}{e_s} \sum_{k=1}^{e_s} \hat{y}_k^i\}_{i=1}^b$;
14:     Obtain prediction loss $l_p$ with $l(\cdot)$ and $\{\hat{y}^i\}_{i=1}^b$;
15:     Obtain subgraph regularization loss $l_o$ with $o(\cdot)$ and $\{\boldsymbol{\alpha}^i\}_{i=1}^b$;
16:     Obtain the final loss $l_f = l_p + \eta \cdot l_o$;
17:     Updated model via backpropagation with $l_f$;
18: **end for**
19: **Return** trained subgraph extraction model $f_c \circ g$;

---

For each input graph along with the label $(G, Y)$, the subgraph extractor $g$ first propagates among $G$ and obtains the node representations $H_i \in \mathbb{R}^h$ for each node. Then, the (edge-centric) sampling attention is obtained as the following

$$\alpha_e = a([H_u, H_v]), \tag{38}$$

for each edge $e = (u, v) \in E$, where $a(\cdot)$ is the attention function and can be simply implemented as a MLP. Note that $\alpha_e$ is slightly different from that in the main text, since we will discuss in detail the discrete sampling process in the implementation.

To enable the gradient backpropagation along with the discrete sampling of subgraphs, we will adopt the Gumbel-softmax trick and straight-through estimator (Jang et al., 2017; Maddison et al., 2017). With the attention from Eq. 38, the sampling probability $\boldsymbol{\beta}$ is then obtained as follows

$$\beta_e = \sigma((\alpha_e + D)/\tau), \tag{39}$$

where $\tau$ is the temperature, $\sigma$ is the sigmoid function, and

$$D = \log U - \log(1 - U),$$

with $U \sim \text{Uniform}(0, 1)$. To sample the discrete subgraph, we sample from the Bernoulli distributions on edges independently

$$A_e \sim \text{Bern}(\beta_e)$$

and obtain the discrete subgraph with each entry as

$$A_e = \text{StopGrad}(A_e - \alpha_e) + \alpha_e, \tag{40}$$

which allows computing the gradients along with the subgraph sampling probability. Although the trick works empirically well, the estimated gradients are approximated ones that have biases from the ground truth. It might be of independent interest to analyze whether the random subgraph sampling in `GMT-sam` can also reduce the gradient estimator biases during discrete sampling.

---

**Algorithm 3** Subgraph classifier training algorithm of **G**raph **M**ultilinear ne**T** (**GMT**).

1: **Input:** Training data $\mathcal{D}_{\text{tr}}$; trained XGNN $f$ with subgraph extractor $g$, and classifier $f_c$ by Alg. 2; length of maximum subgraph classifier training epochs $e_c$; batch size $b$; loss function $l(\cdot)$; subgraph regularization $o(\cdot)$; subgraph regularization weight $\gamma$;
2: Randomly initialize $f$;
3: // Phase of subgraph classifier learning.
4: **for** $j = 1$ to $e_l$ **do**
5:     Sample a batch of data $\{G^i, Y^i\}_{i=1}^b$ from $\mathcal{D}_{\text{tr}}$;
6:     Obtain sampling attention $\{\boldsymbol{\alpha}^i\}_{i=1}^b$ via Eq. 38;
7:     // Soft subgraph propagation.
8:     Obtain edge sampling probability $\{\boldsymbol{\beta}^i = \text{StopGrad}(\boldsymbol{\alpha}^i)\}_{i=1}^b$; // subgraph extractor frozen
9:     Obtain prediction with subgraph $\{\hat{y}^i\}_{i=1}^b$ via weighted message passing with $\{\boldsymbol{\beta}^i\}_{i=1}^b$;
10:     Obtain prediction loss $l_p$ with $l(\cdot)$ and $\{\hat{y}^i\}_{i=1}^b$;
11:     Obtain subgraph regularization loss $l_o$ with $o(\cdot)$ and $\{\boldsymbol{\alpha}^i\}_{i=1}^b$;
12:     Obtain final loss $l_f = l_p + \eta \cdot l_o$;
13:     Updated model via backpropagation with $l_f$;
14: **end for**
15: **Return** final model $f_c \circ g$;

---

**Learning neural subgraph multilinear extension.** When the subgraph extractor is trained, we then enter into stage two, which focuses on extracting the learned subgraph information for better predicting the label with a single pass forward. More concretely, although `GMT` trained with `GMT-sam` improves interpretability, `GMT-sam` still requires multiple random subgraph sampling to approximate `SubMT` and costs much additional overhead. To this end, we propose to learn a neural `SubMT` that only requires a single sampling, based on the trained subgraph extractor $g$ with `GMT-sam`.

Learning the neural `SubMT` is essentially to approximate the MCMC with a neural network, though it is inherently challenging to approximate MCMC (Johndrow et al., 2020; Papamarkou et al., 2022). Nevertheless, the feasibility of neural `SubMT` learning is backed by the inherent causal subgraph assumption of (Chen et al., 2022a), once the causal subgraph is correctly identified, simply learning the statistical correlation between the subgraph and the label is sufficient to recover the causal relation.

Therefore, we propose to simply re-train a new classifier GNN with the frozen subgraph extractor, to distill the knowledge contained in $\widehat{G}_c$ about $Y$. The implementation is simply to stop the gradients of the subgraph extractor, while only optimizing the classifier GNN with the predicted sampling probability. Note that it breaks the shared encoder structure of the `GSAT`, which could avoid potential representation conflicts for a graph encoder shared by both the subgraph extractor and the classifier. Under this consideration, we also enable the BatchNorm (Ioffe & Szegedy, 2015) in the subgraph extractor to keep count of the running stats when training the new classifier.

Empirically, the weighted message passing can effectively capture the desired information from $g$ and lead to a performance increase. This scheme also brings additional benefits over the originally trained classifier, which focuses on providing the gradient guidance for finding proper $G_c$ instead of learning all the available statistical correlations between $G_c$ and $Y$.

### E.2 DISCUSSIONS ON GMT IMPLEMENTATIONS

With the overall algorithm training the subgraph extractor and the classifier, we then discuss in more detail the specific implementation choices of `GMT-sam`.

**Transforming node-centric random subgraph sampling**   In the task of geometric learning, the input graphs are initially represented as point clouds. The graph structures are built upon the node features and geometric knowledge. Therefore, `LRI` adopts the node-centric sampling and learns sampling probabilities for nodes when implementing the graph information bottleneck. However, when sampling concrete subgraphs from a node-centric view, it will often lead to a too aggressive sampling. Otherwise, one has to increase the sampling probability $r$ of the variational distribution $Q(G_c)$ in Eq. 26. To this end, we can transform the node-centric sampling to edge-centric sampling. Let $\alpha_i$ denote the sampling probability for node $i$, then the edge sampling probabilities can be obtained via:

$$\beta_e = \alpha_u \cdot \alpha_v, \tag{41}$$

for each edge $e = (u, v) \in E$. It thus enables the subgraph sampling from the node-centric view. Empirically, in geometric datasets, we observe a lower variance when transforming the node-centric sampling to edge-centric sampling.

**Warmup of `GMT-sam`.**   Although more sampling rounds can improve the approximation precision of `GMT-sam` to `SubMT`, it would also affect the optimization of the interpretable subgraph learning, in addition to the additional unnecessary computational overhead. For example, at the beginning of the interpretable subgraph learning, the subgraph extractor will yield random probabilities like $0.5$.

- First, a more accurate estimation based on random `SubMT` is unnecessary.

- Second, at such random probabilities, every subgraph gets a nearly equal chance of being sampled, and gets gradients backpropagated. Since neural networks are universal approximators, the whole network can easily be misled by the noises, which will slow down the learning speed of the meaningful subgraphs.

- Third, when spurious correlations exist between subgraphs and the labels, the learning process will be more easily misled by the potential spurious correlations at the beginning of the subgraph learning.

More importantly, sampling multiple times can lead to trivial solutions with degenerated performance in the `GSAT` objective. Specifically, the formulation of the mutual information regularizer in `GSAT` has a trivial solution where all $\alpha_e$ directly collapses to the given $r$. More formally, let $\alpha_e = r$ in the following objective obviously lead to zero loss that appears to be a Pareto optimal solution (Chen et al., 2022b) that can be selected as the output:

$$D_{\mathrm{KL}}(\mathrm{Bern}(\alpha_e)||\mathrm{Bern}(r)) = \sum_e \alpha_e \log \frac{r}{r} + (1 - r) \log \frac{(1 - \alpha_e)}{(1 - r)} = 0.$$

The trivial solutions can occur to `GMT` more easily with more rounds of subgraph sampling, especially in too simple or too complicated tasks.

To tackle the above problem, we propose two warmup strategies:

- Larger initial prior $r$ of $Q(G_c)$ in Eq. 26: `GSAT` achieves the objective of graph information bottleneck with a schedule of $r$ in $Q(G_c)$ as $0.9$, which could promote the random sampling probabilities to meaningful subgraph signals. As the random subgraph sampling will slow the optimization, we can warm up the initial subgraph learning with a larger initial $r$. In experiments, we try with $r = 1.0$ and $r = 0.9$, and find $r = 0.1$ can effectively warm up and speed up the subgraph learning, which is especially meaningful for too simple tasks where XGNNs can easily overfit to, or too hard tasks where XGNNs learns the meaningful subgraph signals in a quite slow speed. We can also use a larger regularization penalty at the initial stage to speed up meaningful subgraph learning.

- Single subgraph sampling: As sampling too many subgraphs can bring many drawbacks such as overfitting and slow learning, we propose warm up the initial subgraph learning with a single sampling during the first stage of $r$ (i.e., when $r$ still equals to the initial $r$ in the schedule of `GSAT`). The single subgraph sampling also implicitly promotes meaningful subgraph learning, as it encourages a higher chance even for a small difference in the sampling probability.

In addition to helping with the warmup of the interpretable subgraph, single subgraph sampling also has some additional benefits and effectively tackles the trivial solution of `GSAT` objective. It also brings more variance between meaningful subgraph learning and noisy subgraph learning, and we find using a single random subgraph learning is extremely helpful for simple tasks such as BA_2motifs in our experiments. The implicit variance of single random subgraph sampling also brings additional benefits to maintaining high variance between the signal subgraph and noisy subgraph, which might be of independent interest. It turns out the variance in single subgraph learning can have an implicit regularization preventing the trivial solution.

In experiments, we will use all of the warmup strategies together (i.e., a larger initial $r$, a larger penalty score, and single subgraph sampling) when we observe a performance degeneration in the validation set. Otherwise, we will stick to the original receipt. More details are given in Sec. F.2.

**Sinlge weighted message passing in `GMT-lin`.** Although it has been shown that propagation with the attention only once can effectively reduce the `SubMT` approximation error, it remains unknown which layer the attention should be applied. Empirically, we examine the following three strategies:

- Weighted message passing on the first layer;
- Weighted message passing on the last layer;
- Single weighted message passing of all layers, and then average the logits;

We find applying weighted message passing to the first layer outperforms the other two strategies in experiments, and thus we stick to the first layer weighted message passing scheme. Exploring the reasons behind the intriguing phenomenon will be an interesting future extension.

**Subgraph sampling for neural `SubMT`.** Although the weighted message passing with $\alpha$ produced by the trained subgraph extractor already achieves better performance, it may not maximally extract the full underlying information of the learned subgraph and the labels, since the original function is a MCMC that is not easy to be fitted (Johndrow et al., 2020). Besides, the weighted message passing itself may not be expressive enough due to the expressivity constraints of GNNs (Xu et al., 2019), and also the limitations of the attention-based GNNs (Fountoulakis et al., 2023; Lee et al., 2023).

Therefore, we propose more subgraph sampling strategies along with alternative architecture of the new classifier, in order to best fit the underlying MCMC function. Specifically, we consider the following aspects:

- Initialization: the graph encoder of the new classifier can be initialized from scratch and avoids overfitting, or initialized from the random subgraph sampling trained models;
- Architecture: weighted message passing, or single weighted message passing as that of `GMT-lin`;
- Attention sampling: set the minimum $p\%$ attention scores directly to 0; set the maximum $p\%$ attention scores directly to 1; set the maximum $p\%$ attention scores directly to 1 while set the minimum $(1-p)\%$ attention scores directly to 0;

We examine the aforementioned strategies and choose the one according to the validation performance in experiments. We exhibit the detailed hyperparameter setup in Appendix F.2.

## F  MORE DETAILS ABOUT THE EXPERIMENTS

In this section, we provide more details about the experiments, including the dataset preparation, baseline implementations, models and hyperparameters selection as well as the evaluation protocols.

### F.1  DATASETS

We provide more details about the motivation and construction method of the datasets that are used in our experiments. Statistics of the regular graph datasets are presented in Table 6, and statistics of the geometric graph datasets are presented in Table 7.

Table 6: Information about the datasets used in experiments. The number of nodes and edges are respectively taking average among all graphs.

| DATASETS | # TRAINING | # VALIDATION | # TESTING | # CLASSES | # NODES | # EDGES | METRICS |
|---|---|---|---|---|---|---|---|
| BA-2MOTIFS | 800 | 100 | 100 | 2 | 25 | 50.96 | ACC |
| MUTAG | 2,360 | 591 | 1,015 | 2 | 30.13 | 60.91 | ACC |
| SUPRIOUS-MOTIF $b = 0.5$ | 9,000 | 3,000 | 6,000 | 3 | 45.05 | 65.72 | ACC |
| SUPRIOUS-MOTIF $b = 0.7$ | 9,000 | 3,000 | 6,000 | 3 | 46.36 | 67.10 | ACC |
| SUPRIOUS-MOTIF $b = 0.9$ | 9,000 | 3,000 | 6,000 | 3 | 46.58 | 67.59 | ACC |
| MNIST-75SP | 20,000 | 5,000 | 10,000 | 10 | 70.57 | 590.52 | ACC |
| GRAPH-SST2 | 28,327 | 3,147 | 12,305 | 2 | 10.20 | 18.40 | ACC |
| OGBG-MOLHIV | 32,901 | 4,113 | 4,113 | 2 | 25.51 | 54.94 | AUC |

Table 7: Statistics of the four geometric datasets from Miao et al. (2023).

| | # Classes | # Features in $\mathbf{X}$ | # Dimensions in r | # Samples | Avg. # Points/Sample | Avg. # Important Points/Sample | Class Ratio | Split Scheme | Split Ratio |
|---|---|---|---|---|---|---|---|---|---|
| ActsTrack | 2 | 0 | 3 | 3241 | 109.1 | 22.8 | 39/61 | Random | 70/15/15 |
| Tau3Mu | 2 | 1 | 2 | 129687 | 16.9 | 5.5 | 24/76 | Random | 70/15/15 |
| SynMol | 2 | 1 | 3 | 8663 | 21.9 | 6.6 | 18/82 | Patterns | 78/11/11 |
| PLBind | 2 | 3 | 3 | 10891 | 339.8 | 132.2 | 29/71 | Time | 92/6/2 |

**BA-2Motifs** (Luo et al., 2020) is a synthetic dataset that adopts the Barabási–Albert (BA) graph data model to generate subgraphs in specific shapes. Each graph contains a motif subgraph that is either a five-node cycle or a house. The class labels are determined by the motif, and the motif itself serves as the interpretation of ground truth. The motif is then attached to a large base graph.

**Mutag** (Debnath et al., 1991) is a typical molecular property prediction dataset. The nodes represent atoms and the edges represent chemical bonds. The label of each graph is binary and is determined based on its mutagenic effect. Following Luo et al. (2020); Miao et al. (2022), -NO2 and -NH2 in mutagen graphs are labeled as ground-truth explanations.

**MNIST-sp** (Knyazev et al., 2019) is a graph dataset converted from MNIST dataset via superpixel transformation. The nodes of MNIST-75sp graphs are the superpixels and the edges are generated according to the spatial distance of nodes in the original image. The ground truth explanations of MNIST-75sp are simply the non-zero pixels. As the original digits are hand-written, the interpretation subgraphs could be in varying sizes.

**Suprious-Motif datasets** (Wu et al., 2022b) is a 3-class synthetic datasets based on BA-2Motifs (Ying et al., 2019; Luo et al., 2020) with structural distribution shifts. The model needs to tell which one of three motifs (House, Cycle, Crane) the graph contains. For each dataset, 3000 graphs are generated for each class at the training set, 1000 graphs for each class at the validation set and testing set, respectively. During the construction of the training data, the motif and one of the three base graphs (Tree, Ladder, Wheel) are artificially (spuriously) correlated with a probability of various biases, and equally correlated with the other two. Specifically, given a predefined bias $b$, the probability of a specific motif (e.g., House) and a specific base graph (Tree) will co-occur is $b$ while for the others is $(1 - b)/2$ (e.g., House-Ladder, House-Wheel). The test data does not have spurious correlations with the base graphs, however, test data will use larger base graphs that contain graph size distribution shifts. Following Miao et al. (2022), we select datasets with a bias of $b = 0.5$, $b = 0.7$, and $b = 0.9$. The interpretation ground truth is therefore the motif itself.

**Graph-SST2** (Socher et al., 2013; Yuan et al., 2020b) is converted from a sentiment analysis dataset in texts. Each sentence in SST2 will be converted to a graph. In the converted graph, the nodes are the words and the edges are the relations between different words. Bode features are generated using BERT (Devlin et al., 2019) and the edges are parsed by a Biaffine parser (Gardner et al., 2018). Following previous works (Wu et al., 2022b; Miao et al., 2022; Chen et al., 2022a), our splits are created according to the averaged degrees of each graph. Specifically, we assign the graphs as follows: Those that have smaller or equal to 50-th percentile averaged degree are assigned to training, those that have averaged degree larger than 50-th percentile while smaller than 80-th percentile are assigned to the validation set, and the left are assigned to test set. Since the original dataset does not have the ground truth interpretations, we report only the classification results.

**OGBG-Molhiv** (Hu et al., 2020) is also a molecular property prediction dataset. The nodes represent atoms and the edges represent chemical bonds. The label of each graph is binary and is determined based on whether a molecule inhibits HIV virus replication or not. The training, validation and test

splits are constructed according to the scaffolds (Hu et al., 2020) hence there also exist distribution shifts across different splits. Since the original dataset does not have the ground truth interpretations, we report only the classification results.

In what follows we continue to introduce the four geometric learning datasets. We refer interested readers to Miao et al. (2023) for more details.

**ActsTrack** dataset (Miao et al., 2023):

- Background: **ActsTrack** involves a fundamental resource in High Energy Physics (HEP), employed for the purpose of reconstructing various properties, including the kinematics, of charged particles based on a series of positional measurements obtained from a tracking detector. Within the realm of HEP experimental data analysis, particle tracking is an essential procedure, and it also finds application in medical contexts, such as proton therapy (Schulte et al., 2004). ActsTrack is formulated differently by Miao et al. (2023) from traditional track reconstruction tasks: It requires predicting the existence of a z → μμ decay and using the set of points from the μ's to verify model interpretation, which can be used to reconstruct μ tracks.

- Construction: In the **ActsTrack** dataset, each data point corresponds to a detector hit left by a particle, and it is associated with a 3D coordinate. Notably, the data points in ActsTrack lack any features in the X dimension, necessitating the use of a placeholder feature with all values set to one during model training. Additionally, the dataset provides information about the momenta of particles as measured by the detectors, which has the potential to be employed for assessing fine-grained geometric patterns in the data; however, it is not utilized as part of the model training process. Given that, on average, each particle generates approximately 12 hits, and a model can perform well by capturing the trajectory of any one of the μ (muon) particles resulting from the decay, we report performance metrics in precision@12 following Miao et al. (2023). The dataset was randomly split into training, validation, and test sets, maintaining a distribution ratio of 70% for training, 15% for validation, and 15% for testing.

**Tau3Mu** dataset (Miao et al., 2023):

- Background: **Tau3Mu** involves another application in High Energy Physics (HEP) dedicated to identifying a particularly challenging signature – charged lepton flavor-violating decays, specifically $\tau \to \mu\mu\mu$ decay. This task involves the analysis of simulated muon detector hits resulting from proton-proton collisions. It's worth noting that such decays are heavily suppressed within the framework of the Standard Model (SM) of particle physics (Holstein, 2006), making their detection a strong indicator of physics phenomena beyond the Standard Model (Collaboration, 2020). Unfortunately, $\tau \to \mu\mu\mu$ decay involves particles with extremely low momentum, rendering them technically impossible to trigger using conventional human-engineered algorithms. Consequently, the online detection of these decays necessitates the utilization of advanced models that explore the correlations between signal hits and background hits, particularly in the context of the Large Hadron Collider. Our specific objective is twofold: to predict the occurrence of $\tau \to \mu\mu\mu$ decay and to employ the detector hits generated by the $\mu$ (muon) particles to validate the model's interpretations.

- Construction: **Tau3Mu** uses the data simulated via the PYTHIA generator (Bierlich et al., 2022). The interpretation labels are using the signal sample with the background samples on a per-event basis (per point cloud) while preserving the ground-truth labels. The hits originating from $\mu$ (muon) particles resulting from the $\tau \to \mu\mu\mu$ decay are designated as ground-truth interpretation. The training data only include hits from the initial layer of detectors, ensuring that each sample in the dataset contains a minimum of three detector hits. Each data point in the samples comprises measurements of a local bending angle and a 2D coordinate within the pseudorapidity-azimuth $(\eta - \phi)$ space. Given that, in the most favorable scenario, the model is required to capture hits from each $\mu$ particle, we report precision@3 following Miao et al. (2023). Lastly, the dataset is randomly split into training, validation, and test sets, maintaining a distribution ratio of 70% for training, 15% for validation, and 15% for testing.

**SynMol** dataset (Miao et al., 2023):

- Background: **SynMol** is a molecular property prediction task. While prior research efforts have explored model interpretability within this domain (McCloskey et al., 2018), their emphasis has

been primarily on examining chemical bond graph representations of molecules, often overlooking the consideration of geometric attributes. In our present study, we shift our attention towards 3D representations of molecules. Our specific objective is to predict a property associated with two functional groups, namely carbonyl and unbranched alkane (as defined by McCloskey et al. (2018)), and subsequently employ the atoms within these functional groups to validate our model's interpretations.

- Construction: **SynMol** is constructed based on ZINC (Lin et al., 2022b) following McCloskey et al. (2018) that creates synthetic properties based on the existence of certain functional groups. The labeling criteria involve classifying a molecule as a positive sample if it contains both an unbranched alkane and a carbonyl group. Conversely, molecules lacking this combination are categorized as negative samples. Consequently, the atoms within branched alkanes and carbonyl groups serve as the designated ground-truth interpretation. In addition to specifying a 3D coordinate, each data point within a sample is also associated with a categorical feature signifying the type of atom it represents. While the combined total of atoms in the two functional groups may be limited to just five, it is important to note that certain molecules may contain multiple instances of such functional groups. Consequently, we report precision metric at precision@5 following Miao et al. (2023). Finally, to mitigate dataset bias, the dataset is split into training, validation, and test sets using a distribution strategy following McCloskey et al. (2018); Miao et al. (2023). This approach ensures a uniform distribution of molecules containing or lacking either of these functional groups.

**PLBind** dataset (Miao et al., 2023):

- Background: **PLBind** is to predict protein-ligand binding affinities leveraging the 3D structural information of both proteins and ligands. This task holds paramount significance in the field of drug discovery, as a high affinity between a protein and a ligand is a critical criterion in the drug selection process (Wang & Zhang, 2017; Karimi et al., 2019). The accurate prediction of these affinities using interpretable models serves as a valuable resource for rational drug design and contributes to a deeper comprehension of the underlying biophysical mechanisms governing protein-ligand binding (Du et al., 2016). Our specific mission is to forecast whether the affinity surpasses a predefined threshold, and we achieve this by examining the amino acids situated within the binding site of the test protein to corroborate our model's interpretations.

- Construction: **PLBind** is constructed protein-ligand complexes from PDBind (Liu et al., 2017). PDBind annotates binding affinities for a subset of complexes in the Protein Data Bank (PDB) (Berman et al., 2000), therefore, a threshold on the binding affinity between a pair of protein and ligand can be used to construct a binary classification task. The ground-truth interpretation is then the part of the protein that are within 15A of the ligand to be the binding site (Liu et al., 2022b). Besides, PLBind also includes all atomic contacts (hydrogen bond and hydrophobic contact) for every protein-ligand pair from PDBsum (Laskowski, 2001), where the ground-truth interpretations are the corresponding amino acids in the protein. Every amino acid in a protein is linked to a 3D coordinate, an amino acid type designation, the solvent-accessible surface area (SASA), and the B-factor. Likewise, each atom within a ligand is associated with a 3D coordinate, an atom type classification, and Gasteiger charges. The whole dataset is then partitioned into training, validation, and test sets, adopting a division based on the year of discovery for the complexes, following Stárk et al. (2022).

### F.2 BASELINES AND EVALUATION SETUP

During the experiments, we do not tune the hyperparameters exhaustively while following the common recipes for optimizing GNNs, and also the recommendation setups by previous works. Details are as follows.

**GNN encoder.** For fair comparison, we use the same GNN architecture as graph encoders for all methods, following Miao et al. (2022; 2023). For the backbone of GIN, we use 2-layer GIN (Xu et al., 2019) with Batch Normalization (Ioffe & Szegedy, 2015) between layers, a hidden dimension of 64 and a dropout ratio of 0.3. For the backbone of PNA, we use 4-layer PNA (Corso et al., 2020) with Batch Normalization (Ioffe & Szegedy, 2015) between layers, a hidden dimension of 80 and a dropout ratio of 0.3. The PNA network does not use scalars, while using (mean, min, max, std aggregators. For the backbone of EGNN (Satorras et al., 2021), we use 4-layer EGNN with Batch

Normalization (Ioffe & Szegedy, 2015) between layers, a hidden dimension of $64$ and a dropout ratio of $0.2$. The pooling functions are all sum pooling.

**Dataste Splits.** We follow previous works (Luo et al., 2020; Miao et al., 2022) to split BA-2Motifs randomly into three sets as (80%/10%/10%), Mutag randomly into 80%/20% as train and validation sets where the test data are the mutagen molecules with -NO2 or -NH2. We use the default split for MNIST-75sp given by (Knyazev et al., 2019) with a smaller sampling size following (Miao et al., 2022). We use the default splits for Graph-SST2 (Yuan et al., 2020b), Spurious-Motifs (Wu et al., 2022b) and OGBG-Molhiv (Hu et al., 2020) datasets. For geometric datasets, we use the author provided default splits.

**Baseline implementations.** We use the author provided codes to implement the baselines GSAT (Miao et al., 2022)[4] and LRI (Miao et al., 2023)[5]. We re-run GSAT and LRI under the same environment using the author-recommended hyperparameters for a fair comparison. Specifically, BA-2Motif, Mutag and PLBind use $r = 0.5$, and all other datasets use $r = 0.7$. The $\lambda$ of information regularizer is set to be $1$ for regular graphs, $0.01$ for Tau3Mu, and $0.1$ for ActsTrack, SynMol and PLBind as recommended by the authors. $r$ will initially be set to $0.9$ and gradually decay to the tuned value. We adopt a step decay, where $r$ will decay $0.1$ for every $10$ epochs. As for the implementation of explanation methods, for regular graphs, we directly adopt the results reported. For geometric graphs, we re-run the baselines to obtain the results, as previous results are obtained according to the best validation interpretation performance that may mismatch the practical scenario where the interpretation labels are usually not available.

**Optimization and model selection.** Following previous works, by default, we use Adam optimizer (Kingma & Ba, 2015) with a learning rate of $1e-3$ and a batch size of $128$ for all models at all datasets, except for Spurious-Motif with GIN and PNA, Graph-SST2 with PNA that we will use a learning rate of $3e-3$. When GIN is used as the backbone model, MNIST-75sp is trained for $200$ epochs, and all other datasets are trained for $100$ epochs, as we observe that $100$ epochs are sufficient for convergence at OGBG-Molhiv. When PNA is used, Mutag and Ba-2Motifs are trained for $50$ epochs and all other datasets are trained for $200$ epochs. We report the performance of the epoch that achieves the best validation prediction performance and use the models that achieve such best validation performance as the pre-trained models. All datasets use a batch size of $128$; except for MNIST-75sp with GIN, we use a batch size of $256$ to speed up training due to its large size in the graph setting.

The final model is selected according to the best validation classification performance. We report the mean and standard deviation of $10$ runs with random seeds from $0$ to $9$.

**Implementations of GMT.** For a fair comparison, GMT uses the same GNN architecture for GNN encoders as the baseline methods. We search for the hyperparameters of $r$ from $[r_0 - 0.1, r_0, r_0 + 0.1]$ according to the default $r_0$ given by Miao et al. (2022; 2023). We search the weights of graph information regularizers from $[0.1, 0.5, 1, 2]$ for regular graphs and from $[0.01, 0.1, 1]$ for geometric datasets. To avoid trivial solutions of the subgraph extractor at the early stage, we search for warm-up strategies mentioned in Appendix E.2. Besides, we also search for the decay epochs of the $r$ scheduler to avoid trivial solutions. We search for the sampling rounds from $[1, 20, 40, 80, 100, 200]$ when the memory allows. In experiments, we find GMT already achieves the state-of-the-art results in most of the set-ups without the warm-up. Only in BA-2Motifs and MNIST-75sp with GIN, and in Tau3Mu with EGNN, GMT needs the warmups.

---

[4] https://github.com/Graph-COM/GSAT
[5] https://github.com/Graph-COM/LRI

Table 8: Sensitivity to different subgraph decoding strategies.

| Initialization | Architecture | Attention | Generalization | | | Interpretation | | |
| | | | spmotif-0.5 | spmotif-0.7 | spmotif-0.9 | spmotif-0.5 | spmotif-0.7 | spmotif-0.9 |
|---|---|---|---|---|---|---|---|---|
| | | GSAT | $47.45_{\pm 5.87}$ | $43.57_{\pm 3.05}$ | $45.39_{\pm 5.02}$ | $74.49_{\pm 4.46}$ | $72.95_{\pm 6.40}$ | $65.25_{\pm 4.42}$ |
| new | mul | min0 | $\mathbf{60.09}_{\pm 5.57}$ | $54.34_{\pm 4.04}$ | $\mathbf{55.83}_{\pm 5.68}$ | $85.50_{\pm 2.40}$ | $\mathbf{84.67}_{\pm 2.38}$ | $73.49_{\pm 5.33}$ |
| old | mul | min0 | $58.83_{\pm 7.22}$ | $\mathbf{55.04}_{\pm 4.73}$ | $55.77_{\pm 5.97}$ | $\mathbf{85.52}_{\pm 2.41}$ | $84.65_{\pm 2.42}$ | $73.49_{\pm 5.33}$ |
| new | mul | max1 | $44.49_{\pm 2.65}$ | $49.77_{\pm 2.31}$ | $50.22_{\pm 2.79}$ | $85.50_{\pm 2.39}$ | $84.66_{\pm 2.37}$ | $73.50_{\pm 5.31}$ |
| old | mul | max1 | $45.91_{\pm 2.86}$ | $49.11_{\pm 3.04}$ | $50.30_{\pm 2.07}$ | $85.49_{\pm 2.39}$ | $84.64_{\pm 2.39}$ | $73.50_{\pm 5.35}$ |
| old | mul | min0max1 | $51.21_{\pm 6.46}$ | $50.91_{\pm 6.50}$ | $53.13_{\pm 4.46}$ | $\mathbf{85.52}_{\pm 2.41}$ | $84.66_{\pm 2.43}$ | $73.49_{\pm 5.34}$ |
| new | mul | normal | $47.69_{\pm 5.72}$ | $44.12_{\pm 5.44}$ | $40.69_{\pm 4.84}$ | $84.69_{\pm 2.40}$ | $80.08_{\pm 5.37}$ | $73.48_{\pm 5.34}$ |
| old | mul | normal | $45.36_{\pm 2.65}$ | $44.25_{\pm 5.41}$ | $43.43_{\pm 5.44}$ | $83.52_{\pm 3.41}$ | $80.07_{\pm 5.35}$ | $73.49_{\pm 5.36}$ |
| new | lin | normal | $43.54_{\pm 5.02}$ | $47.59_{\pm 4.78}$ | $46.53_{\pm 3.27}$ | $85.47_{\pm 2.39}$ | $80.07_{\pm 5.37}$ | $\mathbf{73.52}_{\pm 5.34}$ |
| old | lin | normal | $46.18_{\pm 3.03}$ | $46.42_{\pm 5.63}$ | $49.00_{\pm 3.34}$ | $83.51_{\pm 3.39}$ | $80.09_{\pm 5.34}$ | $73.46_{\pm 5.35}$ |

To better extract the subgraph information, we also search for subgraph sampling strategies mentioned in Appendix E.2. Note that the hyperparameter search and training of the classifier is independent of the hyperparameter search of the subgraph extractor. Once could select the best subgraph extractor and train the new classifier onto it. When training the classifier, we search for the following 9 subgraph decoding strategies as shown in Table F.2. Specifically,

- Initialization: "new" refers to that the classifier is initialized from scratch; "old" refers to that the classifier is initialized from the subgraph extractor;
- Architecture: "mul" refers to the default message passing architecture; "lin" refers to the `GMT-lin` architecture;
- Attention: "normal" refers to the default weighted message passing scheme; "min0" refers to setting the minimum $p\%$ attention scores directly to 0; "max0" refers to setting the maximum $p\%$ attention scores directly to 1; "min0max1" refers to setting the maximum $p\%$ attention scores directly to 1 while set the minimum $(1-p)\%$ attention scores directly to 0;

Table F.2 demonstrates the generalization and interpretation performance of `GMT-sam` in spurious motif datasets (Wu et al., 2022b), denoted as "spmotif" with different levels of spurious correlations. It can be found that `GMT-sam` is generically robust to the different choices of the decoding scheme and leads to improvements in terms of OOD generalizability and interpretability.

## F.3 MORE INTERPRETATION RESULTS

We additionally conduct experiments with post-hoc explanation methods based on the PNA backbone. Specifically, we selected two representative post-hoc methods GNNExplainer and PGExplainer, and one representative intrinsic interpretable baseline DIR. The results are given in the table below. It can be found that most of the baselines still significantly underperform GSAT and GMT. One exception is that DIR obtains highly competitive (though unstable) interpretation results in spurious motif datasets, nevertheless, the generalization performance of DIR remains highly degenerated ($53.03_{\pm 8.05}$ on spmotif_0.9).

Table 9: More interpretation results of baselines using PNA

| | BA_2Motifs | Mutag | MNIST-75sp | spmotif_0.5 | spmotif_0.7 | spmotif_0.9 |
|---|---|---|---|---|---|---|
| GNNExp | $54.14_{\pm 3.30}$ | $73.10_{\pm 7.44}$ | $53.91_{\pm 2.67}$ | $59.40_{\pm 3.88}$ | $56.20_{\pm 6.30}$ | $57.39_{\pm 5.95}$ |
| PGE | $48.80_{\pm 14.58}$ | $76.02_{\pm 7.37}$ | $56.61_{\pm 3.38}$ | $59.46_{\pm 1.57}$ | $59.65_{\pm 1.19}$ | $60.57_{\pm 0.85}$ |
| DIR | $72.33_{\pm 23.87}$ | $87.57_{\pm 27.87}$ | $43.12_{\pm 10.07}$ | $85.90_{\pm 2.24}$ | $83.13_{\pm 4.26}$ | $85.10_{\pm 4.15}$ |
| GSAT | $89.35_{\pm 5.41}$ | $99.00_{\pm 0.37}$ | $85.72_{\pm 1.10}$ | $79.84_{\pm 3.21}$ | $79.76_{\pm 3.66}$ | $80.70_{\pm 5.45}$ |
| GMT-lin | $95.79_{\pm 7.30}$ | $99.58_{\pm 0.17}$ | $85.02_{\pm 1.03}$ | $80.19_{\pm 2.22}$ | $84.74_{\pm 1.82}$ | $85.08_{\pm 3.85}$ |
| GMT-sam | $99.60_{\pm 0.48}$ | $99.89_{\pm 0.05}$ | $87.34_{\pm 1.79}$ | $88.27_{\pm 1.71}$ | $86.58_{\pm 1.89}$ | $85.26_{\pm 1.92}$ |

## F.4 COMPUTATIONAL ANALYSIS

We provide more discussion and analysis about the computational overhead required by `GMT`, when compared to `GSAT`. As `GMT-lin` differs only in the number of weighted message passing rounds

from `GSAT`, and has the same number of total message passing rounds, hence `GMT-lin` and `GSAT` have the same time complexity as $O(E)$ for each epoch, or for inference. When comparing `GMT-sam` to `GMT-lin` and `GSAT`, During training, `GMT-sam` needs to process $k$ rounds of random subgraph sampling, resulting in $O(k|E|)$ time complexity; During inference, `GMT-sam` with normal subgraph decoding methods requires the same complexity as `GMT-lin` and `GSAT`, as $O(|E|)$. When with special decoding strategy such as setting part of the attention entries to 1 or 0, `GMT-sam` additionally needs to sort the attention weights, and requires $O(|E| + |E|\log|E|)$ time complexity.

| Training | BA_2Motifs GIN | PNA | MNIST-75sp GIN | PNA | ActsTrack EGNN |
|---|---|---|---|---|---|
| `GSAT` | $0.70_{\pm0.12}$ | $1.00_{\pm0.13}$ | $41.28_{\pm0.61}$ | $80.98_{\pm10.55}$ | $3.57_{\pm1.41}$ |
| `GMT-lin` | $0.68_{\pm0.12}$ | $1.02_{\pm0.15}$ | $41.12_{\pm0.69}$ | $81.11_{\pm10.44}$ | $3.69_{\pm0.93}$ |
| `GMT-sam` | $6.25_{\pm0.48}$ | $17.03_{\pm0.91}$ | $136.60_{\pm1.21}$ | $280.77_{\pm4.00}$ | $5.38_{\pm0.59}$ |
| Inference | | | | | |
| `GSAT` | $0.07_{\pm0.05}$ | $0.11_{\pm0.12}$ | $18.69_{\pm0.35}$ | $24.40_{\pm2.06}$ | $0.84_{\pm0.38}$ |
| `GMT-lin` | $0.08_{\pm0.07}$ | $0.07_{\pm0.01}$ | $18.72_{\pm0.41}$ | $23.81_{\pm1.89}$ | $0.80_{\pm0.21}$ |
| `GMT-sam` (normal) | $0.05_{\pm0.01}$ | $0.12_{\pm0.01}$ | $18.72_{\pm0.35}$ | $18.01_{\pm1.47}$ | $0.50_{\pm0.13}$ |
| `GMT-sam` (sort) | $0.07_{\pm0.01}$ | $0.21_{\pm0.06}$ | $19.07_{\pm0.55}$ | $18.69_{\pm3.35}$ | $0.54_{\pm0.10}$ |

In the table above, we benchmarked the real training/inference time of `GSAT`, `GMT-lin` and `GMT-sam` in different datasets, where each entry demonstrates the time in seconds for one epoch. We benchmark the latency of `GSAT`, `GMT-lin` and `GMT-sam` based on GIN, PNA and EGNN on different scales of datasets. The sampling rounds of `GMT-sam` are set to 20 for PNA on MNIST-sp, 10 for EGNN, and 100 to other setups. From the table, it can be found that, although `GMT-sam` takes longer time for training, but the absolute values are not high even for the largest dataset MNIST-sp. As for inference, `GMT-sam` enjoys a similar latency as others, aligned with our discussion.

### F.5 MORE COUNTERFACTUAL FIDELITY STUDIES

To better understand the results, we provide more counterfactual fidelity results in supplementary to Sec. 3.2 and Fig. 6 and 7. Shown as in Fig. 8, 9, we plot the counterfactual fidelity results of `GSAT` and the simulated `SubMT` via `GMT-sam` with 10 and 100 on BA-2Motifs and Mutag datasets with the distance measure as KL divergence. Fig. 10, 11 show the counterfactual fidelity results of `GSAT` and the simulated `SubMT` via `GMT-sam` with 10 and 100 on BA-2Motifs and Mutag datasets with the distance measure as JSD divergence. It can be found that, the gap in counterfactual fidelity measured in KL divergence or JSD divergence can be even larger between `GSAT` and `SubMT`, growing up to 10 times. These results can serve as strong evidence for the degenerated interpretability caused by the failure of `SubMT` approximation.

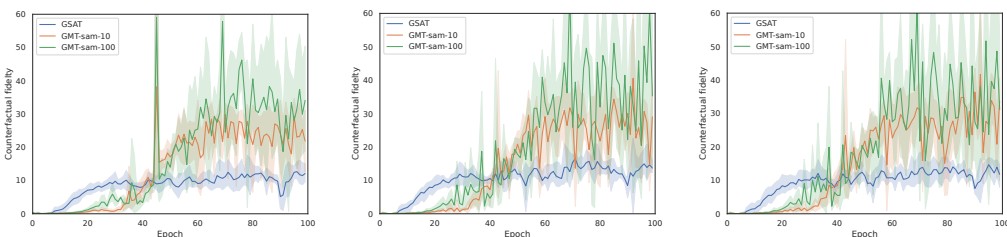

(a) `SubMT` on BA-2Motifs trainset. (b) `SubMT` on BA-2Motifs valset. (c) `SubMT` on BA-2Motifs test set.

Figure 8: Counterfactual fidelity on BA-2Motifs with the distance measure as KL divergence.

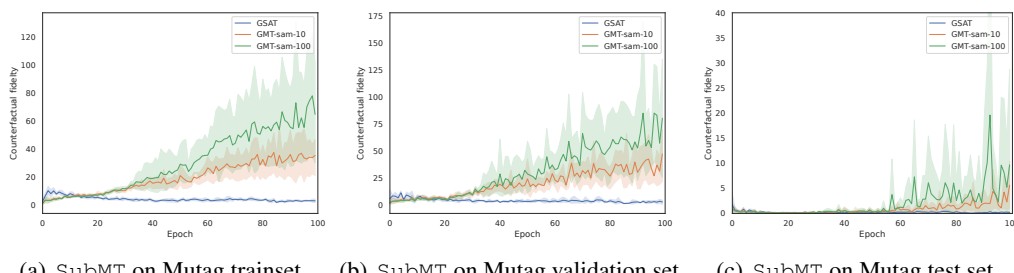

(a) `SubMT` on Mutag trainset. (b) `SubMT` on Mutag validation set. (c) `SubMT` on Mutag test set.

Figure 9: Counterfactual fidelity on Mutag with the distance measure as KL divergence.

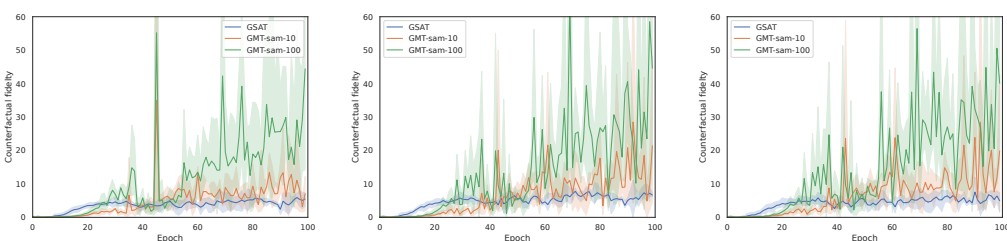

(a) `SubMT` on BA-2Motifs trainset. (b) `SubMT` on BA-2Motifs valset. (c) `SubMT` on BA-2Motifs test set.

Figure 10: Counterfactual fidelity on BA-2Motifs with the distance measure as JSD divergence.

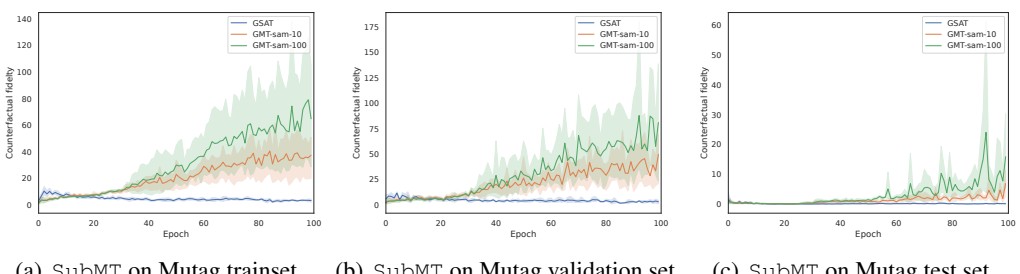

(a) `SubMT` on Mutag trainset. (b) `SubMT` on Mutag validation set. (c) `SubMT` on Mutag test set.

Figure 11: Counterfactual fidelity on Mutag with the distance measure as JSD divergence.

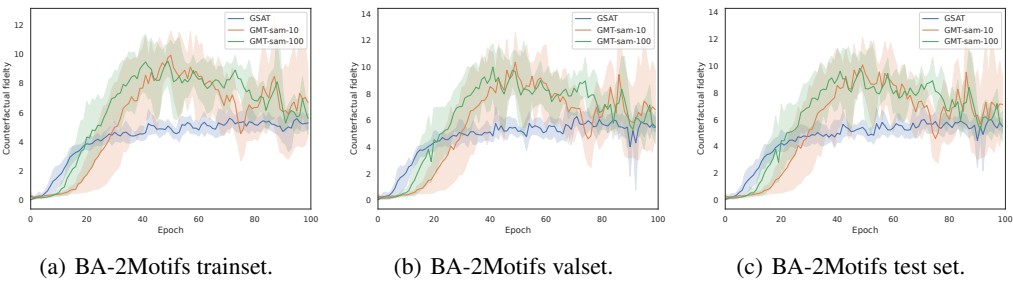

(a) BA-2Motifs trainset. (b) BA-2Motifs valset. (c) BA-2Motifs test set.

Figure 12: The `GMT` optimization issue in terms of counterfactual fidelity on BA-2Motifs.

Shown as in Fig. 12, 13, we plot the counterfactual fidelity results of `GSAT` and the simulated `SubMT` via `GMT-sam` with 10 and 100 on BA-2Motifs and Mutag datasets. Compared to previous results, the `GMT-sam` in Fig. 12, 13 does not use any warmup strategies that may suffer from the optimization

issue as discussed in Sec. E. It can be found that, at the begining of the optimization, `GMT-sam` demonstrates increasing counterfactual fidelity. However, as the optimization keeps proceeding, the counterfactual fidelity of `GMT-sam` will degenerate, because of fitting to the trivial solution of the `GSAT` objective. Consequently, the interpretation results will degenerate too at the end of the optimization.

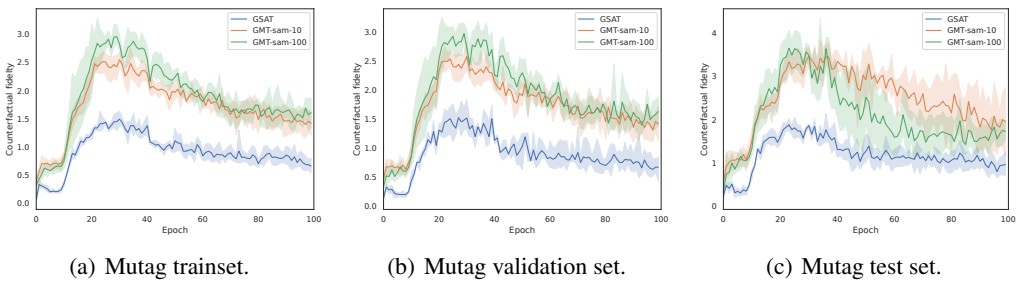

(a) Mutag trainset.    (b) Mutag validation set.    (c) Mutag test set.

Figure 13: The `GMT` optimization issue in terms of counterfactual fidelity on Mutag.

## F.6 SUBMT APPROXIMATION GAP ANALYSIS

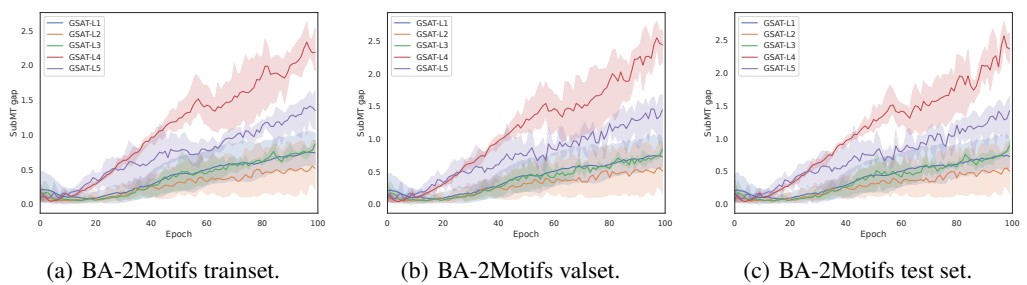

(a) BA-2Motifs trainset.    (b) BA-2Motifs valset.    (c) BA-2Motifs test set.

Figure 14: The `SubMT` approximation gap of `GSAT` with SGC on BA-2Motifs.

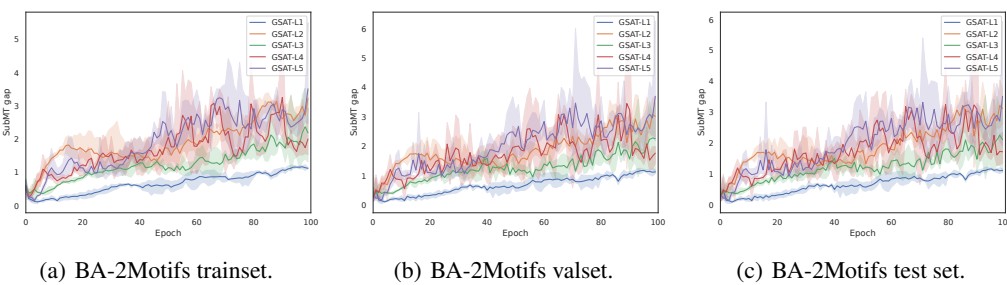

(a) BA-2Motifs trainset.    (b) BA-2Motifs valset.    (c) BA-2Motifs test set.

Figure 15: The `SubMT` approximation gap of `GSAT` with GIN on BA-2Motifs.

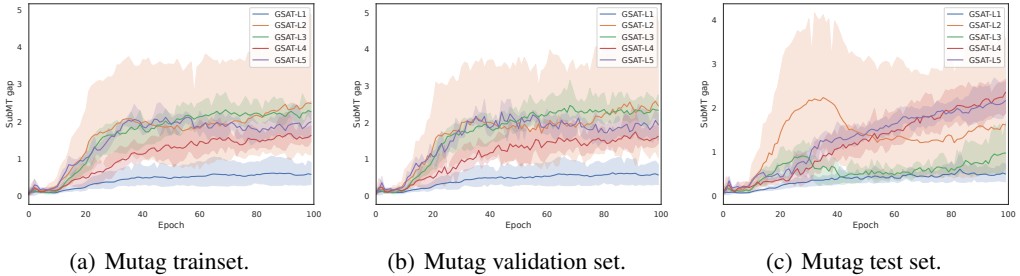

(a) Mutag trainset.     (b) Mutag validation set.     (c) Mutag test set.

Figure 16: The `SubMT` approximation gap of `GSAT` with SGC on Mutag.

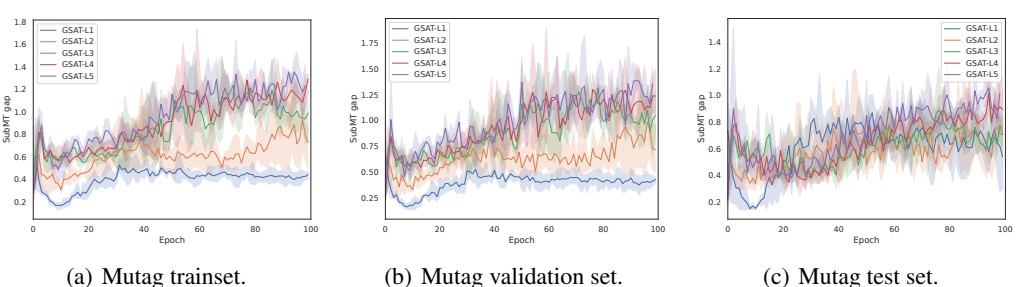

(a) Mutag trainset.     (b) Mutag validation set.     (c) Mutag test set.

Figure 17: The `SubMT` approximation gap of `GSAT` with GIN on Mutag.

Fig. 14 and 15, Fig. 16 and 17 demonstrate the `SubMT` approximation gaps of `GSAT` implemented in GIN and SGC on BA_2Motifs and Mutag respectively. To fully verify Proposition D.4, we range the number of layers of GIN and SGC from 1 to 5. It can be found that the results are well aligned with Proposition D.4. When the number of layers is 1, the `SubMT` approximation gap is smallest, because of more "linearity" in the network. While along with the growing number of GNN layers, the network becomes more "unlinear" such that the `SubMT` approximation gap will be larger.

### F.7 SOFTWARE AND HARDWARE

We implement our methods with PyTorch (Paszke et al., 2019) and PyTorch Geometric (Fey & Lenssen, 2019) 2.0.4. We ran our experiments on Linux Servers installed with V100 graphics cards and CUDA 11.3.

### F.8 INTERPRETATION VISUALIZATION

To better understand the results, we provide visualizations of the learned interpretable subgraphs by `GSAT` and `GMT-sam` in the Spurious-Motif datasets, as well as the learned interpretable subgraphs by `GMT-sam` in OGBG-Molhiv dataset.

The results on Spurious-Motif datasets are given in Fig. 18, 19,20 for $b = 0.5$, $b = 0.7$ and $b = 0.9$, respectively. The red nodes are the ground-truth interpretable subgraphs. It can be found that `GMT-sam` indeed learns the interpretable subgraph better than `GSAT`, which also explains the excellent OOD generalization ability of `GMT-sam` on Spurious Motif datasets.

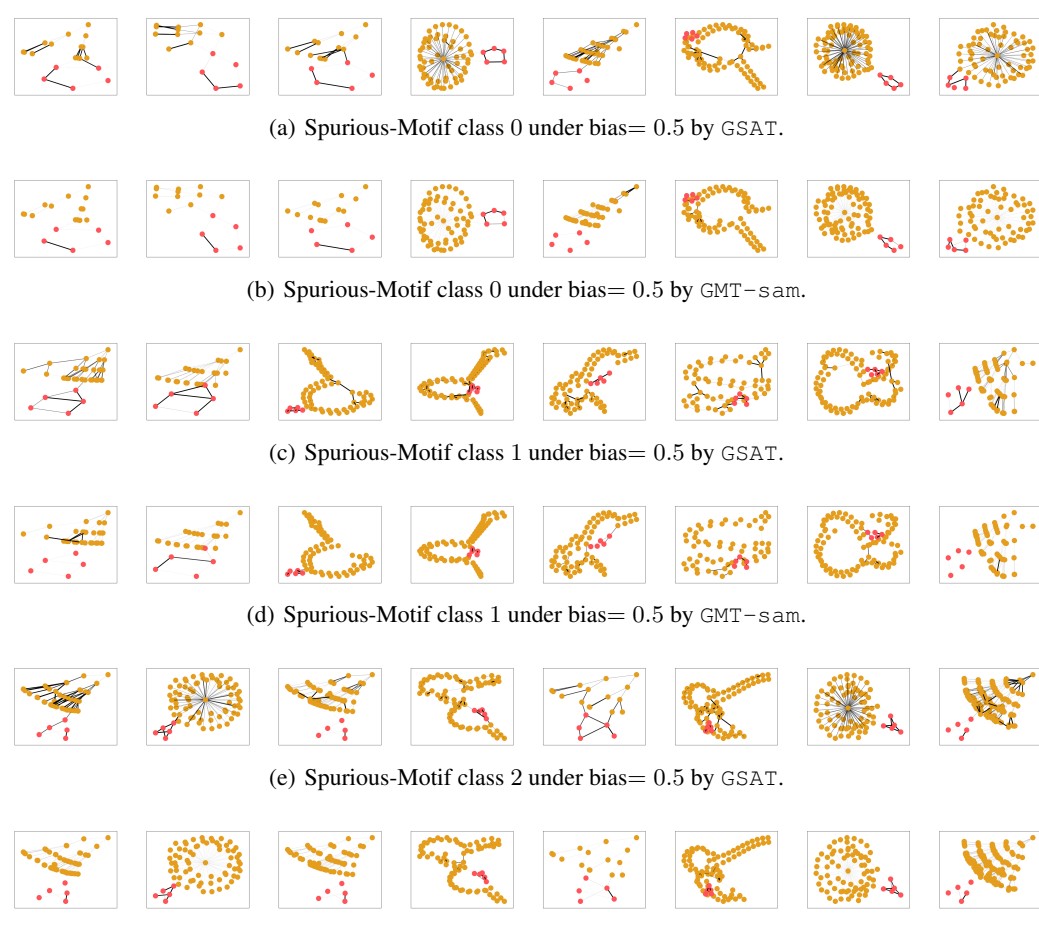

(a) Spurious-Motif class 0 under bias= 0.5 by GSAT.

(b) Spurious-Motif class 0 under bias= 0.5 by GMT-sam.

(c) Spurious-Motif class 1 under bias= 0.5 by GSAT.

(d) Spurious-Motif class 1 under bias= 0.5 by GMT-sam.

(e) Spurious-Motif class 2 under bias= 0.5 by GSAT.

(f) Spurious-Motif class 2 under bias= 0.5 by GMT-sam.

Figure 18: Learned interpretable subgraphs by GSAT and GMT-sam on Spurious-Motif $b = 0.5$.

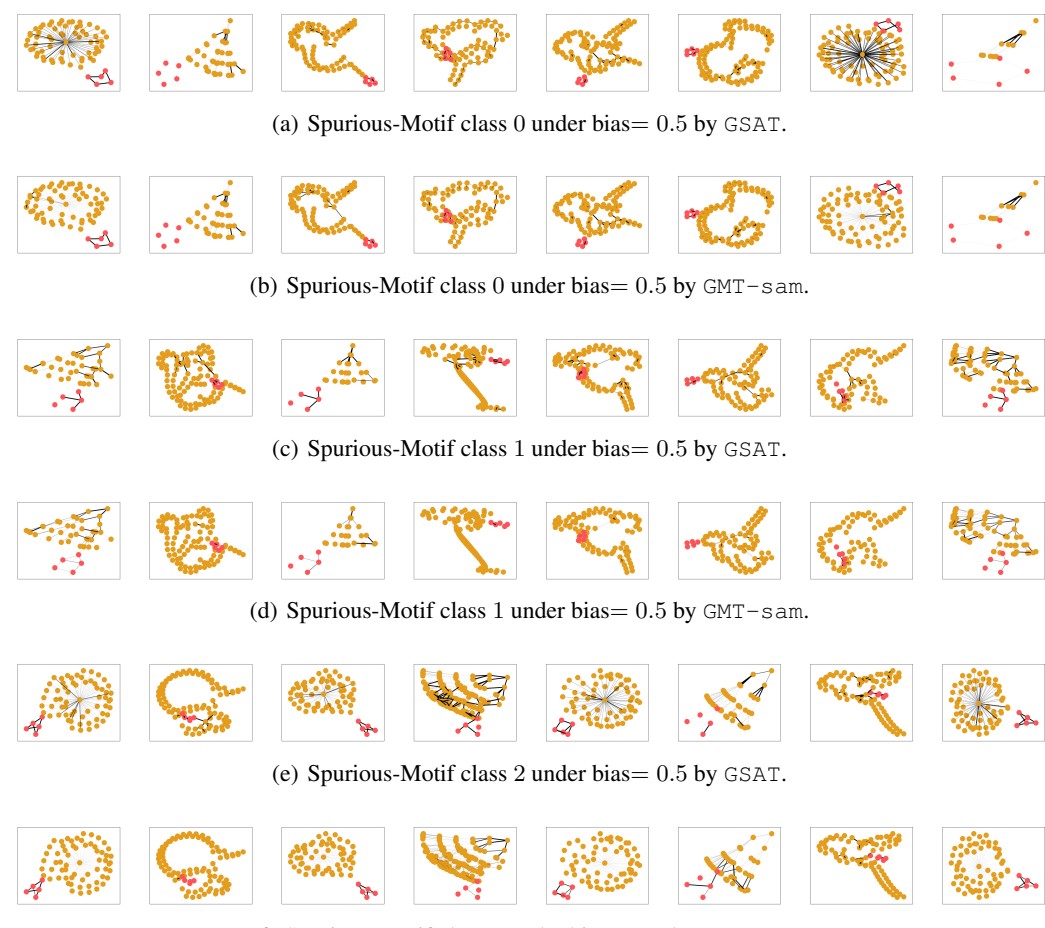

(a) Spurious-Motif class 0 under bias= 0.5 by GSAT.

(b) Spurious-Motif class 0 under bias= 0.5 by GMT-sam.

(c) Spurious-Motif class 1 under bias= 0.5 by GSAT.

(d) Spurious-Motif class 1 under bias= 0.5 by GMT-sam.

(e) Spurious-Motif class 2 under bias= 0.5 by GSAT.

(f) Spurious-Motif class 2 under bias= 0.5 by GMT-sam.

Figure 19: Learned interpretable subgraphs by GSAT and GMT-sam on Spurious-Motif $b = 0.7$.

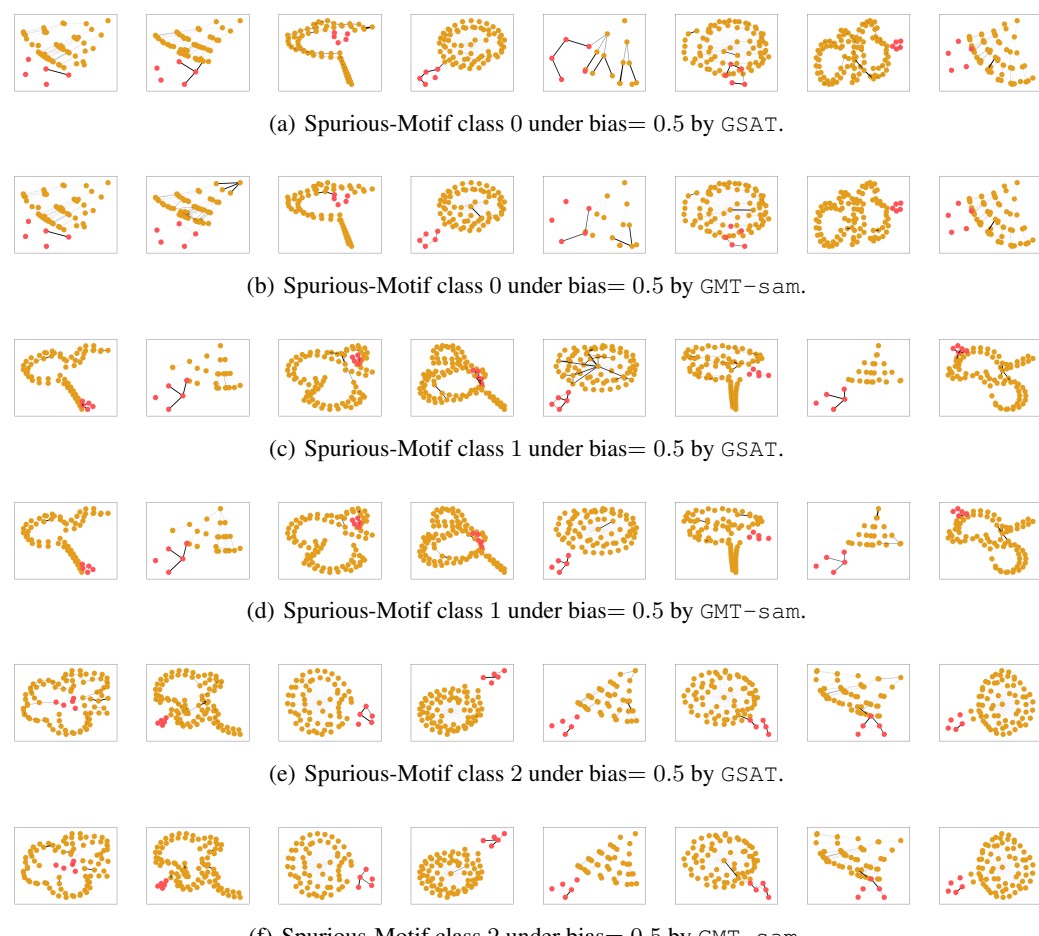

(a) Spurious-Motif class 0 under bias= 0.5 by GSAT.

(b) Spurious-Motif class 0 under bias= 0.5 by GMT-sam.

(c) Spurious-Motif class 1 under bias= 0.5 by GSAT.

(d) Spurious-Motif class 1 under bias= 0.5 by GMT-sam.

(e) Spurious-Motif class 2 under bias= 0.5 by GSAT.

(f) Spurious-Motif class 2 under bias= 0.5 by GMT-sam.

Figure 20: Learned interpretable subgraphs by GSAT and GMT-sam on Spurious-Motif $b = 0.9$.

In addition, we also provide the visualization of interpretable subgraphs learned by GMT-sam on OGBG-Molhiv, given in Fig. 21.

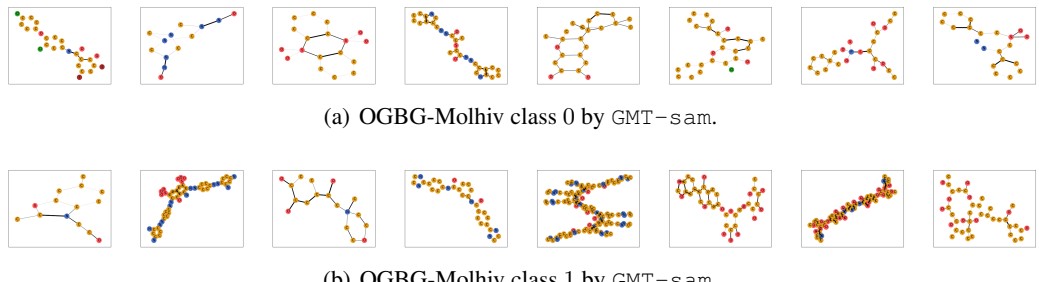

(a) OGBG-Molhiv class 0 by GMT-sam.

(b) OGBG-Molhiv class 1 by GMT-sam.

Figure 21: Learned interpretable subgraphs by GMT-sam on OGBG-Molhiv.

