# OpenReview forum: "Interpretable and Generalizable Graph Neural Networks via Subgraph Multilinear Extension"
_ICLR.cc/2024/Conference — Submitted to ICLR 2024_

### Official Review · Reviewer_uKEu · 2023-11-01

**Soundness:** 3 good
**Presentation:** 3 good
**Contribution:** 3 good
**Rating:** 6
**Confidence:** 3

**Summary:**

In this paper, the authors introduce a self-interpretable graph neural network rooted in the concept of subgraph multilinear extension. The method is elaborated from a perspective of counterfactual fidelity, and the authors put forward both linear and sampling-based graph neural networks (GNNs) for subgraph extraction. The experimental results suggest that the proposed method is effective in achieving its intended goals.

**Strengths:**

1. The proposed method is well-supported by theoretical foundations. The authors provide an insightful explanation of the causal subgraph from a fidelity perspective and derive a method that aligns well with this theoretical underpinning.

2. The authors offer a theoretical framework for improving graph interpretability and shed light on why existing methods often fall short in this regard. They also articulate how they aim to meet this criterion and present experimental results that appear to support their proposition.

**Weaknesses:**

The authors conduct multiple comparisons between the proposed method and GSAT throughout the paper. However, it appears that the primary distinction lies in the sampling process, with the proposed method incorporating a higher number of samples than previously employed in GSAT. It would be valuable for the authors to emphasize this key differentiator and provide a more detailed discussion regarding the impact of increased sampling on the method's performance and results. This would enable a clearer understanding of the specific advantages of their approach over GSAT.

**Questions:**

For figure 2(b) and 2(c), what is the distance metric specifically used? Will there be any difference in the results using different metrics?

---

> ### Author Response · Authors · 2023-11-18
> **Response to Reviewer uKEu [1/1]**
>
> Thank you for your insightful comments on our work. Please see our responses to your comments and suggestions below.
>
> > W1.1. It appears that the primary distinction lies in the sampling process, with the proposed method incorporating a higher number of samples than previously employed in GSAT.
>
> We’d like to clarify that, technically, there are two key innovations in GMT compared to GSAT:
> - The first one is the accurate estimation of SubMT, either through the linearized variant (GMT-LIN) or the multiple random subgraph sampling variant (GMT-sam). Note that the weighted message passing scheme in GSAT cannot accept discrete input, and will suffer from the biases from both the approximation of SubMT and the Gumbel softmax trick[1,2]. GMT-sam overcomes the two drawbacks in GSAT simultaneously with the random subgraph sampling scheme implemented via straight-through estimator.
> - The second is the efficient approximation of the neural subgraph multilinear extension, which is typically non-trivial as it requires the MCMC sampling. We thus develop multiple subgraph decoding strategies to efficiently decode from the neural subgraph multlinear extension, and reduce the inference latency competitively to that of a vanilla GNN. The second innovation is critical to the deployment of GMT-sam in many realistic applications.
> We have revised our manuscript to highlight the key differentiators.
>
> > W1.2. It would be valuable for the authors to emphasize this key differentiator and provide a more detailed discussion regarding the impact of increased sampling on the method's performance and results.
>
> Thank you for the constructive suggestion! We have revised our manuscript to include the following discussion:
> - Theoretically, as shown in Theorem 5.1, increasing the sampling grounds in GMT-sam could result in a more accurate approximation of the SubMT. However, it can also bring drawbacks to the optimization with the increased sampling.
> - On the one hand, it requires unnecessary computational overhead with a large number of sampling. From Figure 3 (b) and (c), it can be found that the performance will saturate after a sufficient number of samplings such as 80 or 100. Then, it is unnecessary to increase the sampling rounds, since a smaller number of sampling rounds is already sufficient to guide the model to find the causal subgraph for the task.
> - On the other hand, in some cases where the task is too simple or too complicated, as discussed in Appendix E.2, a large number of sampling rounds at the beginning can lead the model more easily fit to noises, spurious correlations, or even a trivial solution of the GSAT objective. In this case, we find that sampling only once could benefit the optimization similarly to the warmup strategy.
>
> > Q1. For Figures 2(b) and 2(c), what is the distance metric specifically used? Will there be any difference in the results using different metrics?
>
> Throughout the paper, our discussion and experiments adopt the total variation distance as the distance metric. Following your suggestion, we also conducted extensive experiments with KL divergence and JSD divergence. As shown in Figures 8,9,10,11 in the revised version, GMT achieves even more improvements up to 10 times larger than that of GSAT when measured in KL or JSD divergence. The results align with those using the total variation distance and demonstrate the generality of our theory and method.
>
> Please let us know if you have any further questions. We’d sincerely appreciate it if you could take the above responses into consideration when making the final evaluation of our work!
>
> **References**
>
> [1] Categorical reparameterization with gumbel-softmax, ICLR 2017.
>
> [2] The concrete distribution: A continuous relaxation of discrete random variables, ICLR 2017.

---

> ### Author Response · Authors · 2023-11-20
> **A summary of response to Reviewer uKEu**
>
> Dear Reviewer uKEu,
>
> We once again thank you for your valuable time and constructive comments in reviewing our work. We have replied to each of the weaknesses/questions raised in your review. To summarize,
> - We discussed more clearly the techical contributions of GMT-sam compared to GSAT, and how to properly tune the sampling rounds;
> - We evaluated the counterfactual fidelity with other distance metrics such as KL divergence, and found GMT-sam can achieve even better results;
>
> Please let us know if there are any of your concerns have not been addressed in our response. We would sincerely appreciate it if you jointly could take our response into consideration when making the final evaluation of our work!

---

> > ### Comment · Reviewer_uKEu · 2023-11-21
> > **Another concern for the baseline results**
> >
> > I appreciate the authors' response, which has addressed several of my previous questions.
> >
> > However, upon revisiting the original GSAT paper, I have a new concern. It appears that the previous baseline results (from GNNEXPLAINER to DIR in Table 1) used for comparison in this paper are identical to those reported in GSAT. Nevertheless, the reported results of GSAT in some cases exhibit deviations from those in its original paper. Moreover, in certain datasets, the deviations are even more substantial (e.g., SPURIOUS-MOTIF and MNIST-75SP). Could the authors provide an explanation for why the reported results of the previous baselines are consistent with those in GSAT while there are deviations in the GSAT paper?
> >
> > Thank you!

---

> ### Author Response · Authors · 2023-11-21
> **A gentle reminder for the closing rebuttal window**
>
> Dear Reviewer uKEu,
>
> We would like to remind you that the rebuttal will be closed within 48 hours. To allow us sufficient time to discuss with you any of your remaining concerns, we would appreciate it if you could take some time to read our rebuttal and give us some feedback. Thank you very much.

---

> ### Comment · Area_Chair_9YMh · 2023-11-21
>
> Dear reviewer,
>
> It is customary to respond to the rebuttal, even if it does not change your mind. Have the authors convinced you with their rebuttal or are you sticking to your score?
>
> Thanks!

---

> ### Author Response · Authors · 2023-11-22
> **Explanation for the baseline results**
>
> Thanks for your reply and careful observation. We are happy to learn that your previous concerns have been addressed. We are confident to address the remaining concern. Please find our answer in the below:
> - The main reason for the performance deviations of GSAT from the original work is that the original GSAT performances are obtained via a **mathematically incorrect implementation**, as acknowledged by the authors ([link](https://github.com/Graph-COM/GSAT/issues/10)). Nevertheless, as explained by the authors, in their follow-up work LRI, they found that mathematically correct implementation could have a better performance.
> - **To ensure a mathematically consistent and fair comparison, we implement all methods strictly following the mathematically correct formulations in all experimental settings**. Nevertheless, GMT-sam can still outperform GSAT in most of the settings regardless of which form GSAT is implemented. We showcase via spurious-motif datasets:
>
> |                       | Generalization  |                 |                 | Interpretation  |                 |                 |
> |-----------------------|-----------------|-----------------|-----------------|-----------------|-----------------|-----------------|
> |                       | spmotif-0.5     | spmotif-0.7     | spmotif-0.9     | spmotif-0.5     | spmotif-0.7     | spmotif-0.9     |
> | GSAT (math correct)   | 47.45+-5.87     | 43.57+-3.05     | 45.39+-5.02     | 74.49+-4.46     | 72.95+-6.40     | 65.25+-4.42     |
> | GSAT (math incorrect) | 52.74+-4.08     | 49.12+-3.29     | 44.22+-5.57     | 78.45+-3.12     | 74.07+-5.28     | 71.97+-4.41     |
> | GMT-sam               | **60.09+-5.57** | **54.34+-4.04** | **55.83+-5.68** | **85.50+-2.40** | **84.67+-2.38** | **73.49+-5.33** |
>
>
> - As for the other baselines, they do not have the issue, while having a significant performance gap from GSAT. Therefore we directly adopt the reported results in the paper of GSAT.
>
>
> Please let us know any of your remaining concerns. Thank you!

---

> > ### Comment · Reviewer_uKEu · 2023-11-23
> >
> > Thank the authors for their explanation. I will take into consideration their experiments in the final decision.

---

> > > ### Author Response · Authors · 2023-11-23
> > > **Thank you**
> > >
> > > Thank you for the follow-up reply. Please let us know if there's anything you feel is not clear and would like us to further clarify. We would like to express our sincere gratitude for your time and constructive comments in reviewing our work!

---

### Official Review · Reviewer_pmQk · 2023-11-01

**Soundness:** 3 good
**Presentation:** 3 good
**Contribution:** 3 good
**Rating:** 8
**Confidence:** 2

**Summary:**

The paper seeks a theoretical understanding of the representational properties and limitations of interpretable GNNs (XGNNs). The paper identifies the ability to approximate the so-called Subgraph Multilinear Extension as the key distribution for interpretable subgraph learning. Building on this observation, they propose the GMT model which implements such an approximation. The paper experimentally demonstrates the superiority of GMT in interpretation and generalization.

**Strengths:**

The paper is overall very well-written and the ideas are neatly structured. The authors are able to devise a theoretical framework for characterizing the expressivity of XGNNs, and shows that the existing XGNNs fail to approximate the subgraph multilinear extension function. The proposed XGNN admits a natural design in light of the above deficiency. The experimental protocol is quite thorough and the conducted experiments adequately test the hypothesis.

**Weaknesses:**

The prediction performance of the proposed model does not see too much improvement over the state-of-the-art. There is not much discussion over the time complexity of GMT-SAM version, vis-a-vis the GMT-LIN version.

**Questions:**

1. Page 4: "Note that $f_c$ is a GNN defined only for discrete graph-structured inputs (i.e., $α \in [0, 1]^m$)". Do you mean all Boolean m-length vectors, because otherwise this is not really discrete.

2. Page 4: "which implicitly assumes the random graph data model (Erdos & Renyi, 1984). Def. 3.1 can also be generalized to other graph models with the corresponding parameterization ... ". Can you please comment on this further if this admits such a straightforward generalization to other models than ER, such SBMs or graphons as you suggested?

3. What is $k$ in Eq. 9? Does it mean an application of k-layers of linear message-passing layers? In continuation, in Eq 11, what is the intuition behind the Hadamard operation between $\hat{A}$ and $A^{k-1}$? Section 5.1 needs to be written more clearly.

---

> ### Author Response · Authors · 2023-11-18
> **Response to Reviewer pmQk [1/2]**
>
> Thank you for your constructive comments. We hope our responses to your comments and suggestions below could make you more confident in supporting our work!
>
> > W1.1. The prediction performance of the proposed model does not see too much improvement over the state-of-the-art.
>
> We’d like to clarify that the generalization performance of the network also depends on the objective itself. In this work, we focus on the graph information bottleneck objective as GSAT[1] that can generalize under certain distribution shifts (i.e., FIIF shifts)[2].
> - For real-world datasets, it’s unknown for the types of distribution shifts, thus the improvements to generalization performance may be limited.
> - For spurious motif datasets, they are generated exactly following the FIIF shifts, hence **it’s expected that the graph information bottleneck shall hold stable and high generalization performance on spurious motif datasets**. From the results it can be found that GSAT fails to do so as GSAT can not reliably approximate SubMT, while GMT yields a stable and good generalization performance, improving GSAT up to 10% in prediction performance.
>
> > W1.2. There is not much discussion over the time complexity of GMT-sam version, vis-a-vis the GMT-LIN version.
>
> We have supplemented the discussion regarding the time complexity of GMT variants as well as GSAT in the revised version. Our discussion focuses on vanilla GNNs implemented in message passing scheme and one could generalize to other more complicated GNNs by correspondingly incorporating their message passing complexity.
> - Let $E$ be the number of edges for the graph. As GMT-LIN differs only in the number of weighted message passing rounds from GSAT, and has the same number of total message passing rounds, hence GMT-LIN and GSAT have the same time complexity as $O(E)$ for each epoch, or for inference.
>
> When comparing GMT-sam to GMT-LIN and GSAT,
> - During training, GMT-sam needs to process $k$ rounds of random subgraph sampling, resulting in $O(kE)$ time complexity;
> - During inference, GMT-sam with normal subgraph decoding methods requires the same complexity as GMT-LIN and GSAT, as $O(E)$.
> When with special decoding strategy such as setting part of the attention entries to $1$ or $0$,  GMT-sam additionally needs to sort the attention weights, and requires $O(E+E\log E)$ time complexity;
>
> In the table below, we benchmarked the real training/inference time of GSAT, GMT-LIN and GMT-sam in different datasets, where each entry demonstrates the time in seconds for one epoch. We benchmark the latency of GSAT, GMT-LIN and GMT-sam based on GIN, PNA and EGNN on different scales of datasets. The sampling rounds of GMT-sam are set to 20 for PNA on MNIST-75sp, 10 for EGNN, and 100 to other setups.
> |                  | BA_2Motifs |             | MNIST-75sp     |              | ActsTrack  |
> |------------------|------------|-------------|--------------|--------------|------------|
> | Training         | GIN        | PNA         | GIN          | PNA          | EGNN       |
> | GSAT             | 0.70+-0.12 | 1.00+-0.13  | 41.28+-0.61  | 80.98+-10.55 | 3.57+-1.41 |
> | GMT-LIN          | 0.68+-0.12 | 1.02+-0.15  | 41.12+-0.69  | 81.11+-10.44 | 3.69+-0.93 |
> | GMT-sam          | 6.25+-0.48 | 17.03+-0.91 | 136.60+-1.21 | 280.77+-4.00 | 5.38+-0.59 |
> | Inference        |            |             |              |              |            |
> | GSAT             | 0.07+-0.05 | 0.11+-0.12  | 18.69+-0.35  | 24.40+-2.06  | 0.84+-0.38 |
> | GMT-LIN          | 0.08+-0.07 | 0.07+-0.01  | 18.72+-0.41  | 23.81+-1.89  | 0.80+-0.21 |
> | GMT-sam (normal) | 0.05+-0.01 | 0.12+-0.01  | 18.72+-0.35  | 18.01+-1.47  | 0.50+-0.13 |
> | GMT-sam (sort)   | 0.07+-0.01 | 0.21+-0.06  | 19.07+-0.55  | 18.69+-3.35  | 0.54+-0.10 |
>
> From the table, it can be found that,
> - Although GMT-sam takes a longer time for training, but the absolute values are not high even for the largest dataset MNIST-75sp. When compared to other intrinsic interpretable methods, GMT-sam consumes a similar training time around 6 hours on MNIST-75sp as DIR.
> - As for inference, GMT-sam enjoys a similar latency as others, aligned with our discussion.

---

> ### Author Response · Authors · 2023-11-18
> **Response to Reviewer pmQk [2/2]**
>
> > Q1. Do you mean all Boolean m-length vectors, because otherwise this is not really discrete.
>
> Yes, we have revised the description to $\boldsymbol{\alpha}\in\\{0,1\\}^m$ to avoid any confusion.
>
> > Q2. Can you please comment on this further if this admits such a straightforward generalization to other models than ER, such SBMs or graphons as you suggested?
>
> One of the key differences between ER and the other two graph models SBMs and Graphons is that, the latter considers the generation of the graph in a blockwise manner. Within each block, the edges have a similar probability of occurring among nodes in the block. The subgraph multilinear can therefore be extended to a hierarchical variant to adhere to the blockwise property.
>
> > Q3.1. What is $k$ in Eq 9? Does it mean an application of k-layers of linear message-passing layers?
>
> Yes, $k$ in Eq 9 refers to the $k$ layers of message passing. We have supplemented the explanation to Eq 9 in our revised version.
>
> > Q3.2. What is the intuition behind the Hadamard operation in Eq 11? Section 5.1 needs to be written more clearly.
>
> As the non-linearity of $k>1$ weighted message passing in Eq 9 is the main cause for the approximation failure, the key idea for adopting the Hardamoard operation is to inject more “linearity” in the message passing by reducing the weighted message passing rounds to $1$. We revised Section 5.1 accordingly to improve its clarity.
>
> Please let us know if you have any further questions. We’d be grateful if you could take the above responses into consideration when making the final evaluation of our work!
>
> **References**
>
> [1] Interpretable and Generalizable Graph Learning via Stochastic Attention Mechanism, ICML 2022.
>
> [2] Learning Causally Invariant Representations for Out-of-Distribution Generalization on Graphs, NeurIPS 2022.

---

> > ### Comment · Reviewer_pmQk · 2023-11-19
> >
> > Thanks for the reply. I intend to maintain my rating for the paper.

---

> > > ### Author Response · Authors · 2023-11-20
> > > **Thank you**
> > >
> > > Thank you for your reply! We hope our response before could make you more confident in supporting our work!

---

### Official Review · Reviewer_gapk · 2023-11-04

**Soundness:** 3 good
**Presentation:** 2 fair
**Contribution:** 3 good
**Rating:** 6
**Confidence:** 3

**Summary:**

This paper focuses on investigating theoretical foundations for interpretable graph neural networks (XGNN). To study the expressive power of interpretable GNNs, the authors propose a framework that formulates interpretable subgraph learning with the multilinear extension of the subgraph predictivity (subMT). To obtain a more accurate approximation of SubMT, the authors design an XGNN architecture (GMT). The superior empirical results on several graph classification benchmarks support the theoretical findings.

**Strengths:**

1. It's interesting to see some works focusing on better understanding the theoretical foundation for the expressive power of XGNN. The motivation to consider the connection between interpretable subgraph learning and multilinear extension is also sound.
2. There are some empirical results to directly support the theoretical findings/claims, which have some merits. For example, the ablation studies demonstrate a better performance on counterfactual fidelity which supports the claims in Sec 4.2.
3. The overall empirical performance of the proposed method seems significantly better than those of baselines.

**Weaknesses:**

1. For Proposition 3.3, in my understanding, it basically says with > 2-layer linear GNN, Eq (6) cannot approximate SubMT. Do you provide any proof of that? Besides, I am curious if we have any empirical findings regarding this.
2. For interpretation performance comparison, the authors provide more baseline methods for GIN while PNA only has GSAT. I wonder what the reason is for this. Could you provide more baselines with PNA as the backbone?

**Questions:**

See the above questions in weaknesses.
Minors:
1. Table 4 is before Table 2 and 3. It would be better to fix this.

---

> ### Author Response · Authors · 2023-11-18
> **Response to Reviewer gapk [1/1]**
>
> Thank you for your insightful comments on our work! Please find our detailed responses to your comments and suggestions below.
>
> > W1.1. Do you provide any proof and empirical findings for Proposition 3.3?
>
> Thank you for pointing out the missed proof of Proposition 3.3. We have supplemented the proof in the revised version. The idea to prove for linear GNN is intuitive:
> a. First, one could write the outcomes of $f\_c(A)=\rho(A^kXW)$ as a summation in $A^k\_{i,j}v\_j$, with $v\_j$ is the j-th entry of the vector $v=XW$.
>
> b. Given the linearity of expectations, the comparison between $E[f_c(A)]$ and $f_c(E[A])$ now turns into the comparison between $E[A^k_{i,j}v_j]$ and $(E[A_{i,j}])^kv_j$.
>
> c. Since $A_ij$ is drawn from the Bernoulli distribution, with the expectation as $\widehat{A}\_{i,j}$, it suffices to know that $E[A^k\_{i,j}v\_j]=1^k\widehat{A}_{i,j}+0^k(1-\widehat{A}\_{i,j})=\widehat{A}\_{i,j}$, while $(E[A\_{i,j}])^k=\widehat{A}\_{i,j}^k$. Then, we know that $E[f\_c(A)] \neq f\_c(E[A])$.
>
> > W1.2. Do you provide any empirical findings for Proposition 3.3?
>
> Yes. We have conducted experiments to measure the total variation distance between the simulated SubMT, i.e., $E[f_c(A)]$, and the previous weighted message passing, i.e., $f_c(E[A])$.  Shown as in Fig. 14 to 17 in the revised version, we evaluate the SubMT approximation gaps of GSAT implemented in GIN and SGC on BA\_2Motifs and Mutag respectively. To fully verify Proposition 3.3, we range the number of layers of GIN and SGC from 1 to 5. It can be found that the results are well aligned with Proposition 3.3:
> - When the number of layers is 1, the SubMT approximation gap is generically the smallest, because of more ``linearity'' in the network.
> -  While along with the growing number of GNN layers, the network becomes more ``unlinear'' such that the SubMT approximation gap will be larger.
>
> > W2.1. For interpretation performance comparison, why do the authors provide more baseline methods for GIN while PNA only has GSAT?
>
> We follow the GSAT experimental setting to conduct the comparison [1]. Since it has been theoretically shown that post-hoc methods are suboptimal in generating the explanation and sensitive to the pre-trained models. The empirical results with GIN can already verify the results.
>
> With the results, and given the fact that GSAT has achieved significant improvements over the previous baselines, we thus compare with the other baselines only based on the GIN backbone.
>
> > W2.2. Could you provide more interpretation baselines with PNA as the backbone?
>
> Yes. We have conducted experiments with post-hoc explanation methods based on the PNA backbone. Specifically, we selected two representative post-hoc methods GNNExplainer and PGExplainer, and one representative intrinsic interpretable baseline DIR. The results are given in the table below. It can be found that most of the baselines still significantly underperform GSAT and GMT. One exception is that DIR obtains highly competitive (though unstable) interpretation results in spurious motif datasets, nevertheless, the generalization performance of DIR remains highly degenerated (53.03±8.05 on spmotif_0.9).
> |         | BA_2Motifs     | Mutag          | MNIST-75sp     | spmotif_0.5    | spmotif_0.7    | spmotif_0.9    |
> |---------|----------------|----------------|----------------|----------------|----------------|----------------|
> | GNNExp  | 54.14±3.30     | 73.10±7.44     | 53.91±2.67     | 59.40±3.88     | 56.20±6.30     | 57.39±5.95     |
> | PGE     | 48.80±14.58    | 76.02±7.37     | 56.61±3.38     | 59.46±1.57     | 59.65±1.19     | 60.57±0.85     |
> | DIR     | 72.33±23.87    | 87.57±27.87    | 43.12±10.07    | 85.90±2.24     | 83.13±4.26     | 85.10±4.15     |
> | GSAT    | 89.35±5.41     | 99.00±0.37     | 85.72±1.10     | 79.84±3.21     | 79.76±3.66     | 80.70±5.45     |
> | GMT-lin | 95.79±7.30     | 99.58±0.17     | 85.02±1.03     | 80.19±2.22     | 84.74±1.82     | 85.08±3.85     |
> | GMT-sam | **99.60±0.48** | **99.89±0.05** | **87.34±1.79** | **88.27±1.71** | **86.58±1.89** | **85.26±1.92** |
>
>
> > Q1. Positions of Table 2-4.
>
> We have fixed it in the revised version.
>
> Please let us know if you have any further questions. We’d be grateful if you could take the above responses into consideration when making the final evaluation of our work!
>
> **References**
>
> [1] Interpretable and Generalizable Graph Learning via Stochastic Attention Mechanism, ICML 2022.

---

> > ### Comment · Reviewer_gapk · 2023-11-23
> >
> > Thanks for the detailed rebuttal. The response seems sound to me especially the proof and the empirical evidence for Proposition 3.3. I keep my score as 6.

---

> > > ### Author Response · Authors · 2023-11-23
> > > **Thank you**
> > >
> > > Thank you for acknowledging the soundness of our response. We would like to express our sincere gratitude for your time and constructive comments in reviewing our work!

---

> ### Author Response · Authors · 2023-11-20
> **A summary of response to Reviewer gapk**
>
> Dear Reviewer gapk,
>
> We again thank you for your valuable time and constructive comments in reviewing our work. We have responded to each of your concerns. In summary,
> - We provided both theoretical and empirical evidence for Proposition 3.3;
> - We conducted extensive experiments with more interpretation baselines using PNA, where the results demonstrate the consistent improvements of GMT over previous methods;
> - We also revised and updated our manuscript following your suggestions;
>
> Please let us know if you have any further questions. We’d sincerely appreciate it if you could take our previous responses into consideration when making the final evaluation of our work!

---

> ### Author Response · Authors · 2023-11-21
> **A gentle reminder for the closing rebuttal window**
>
> Dear Reviewer gapk,
>
> We would like to remind you that the rebuttal will be closed very soon. To allow us sufficient time to discuss with you any of your remaining concerns, we would appreciate it if you could take some time to read our rebuttal and give us some feedback. Thank you very much.

---

> ### Author Response · Authors · 2023-11-22
> **The rebuttal window will be closed very soon**
>
> Dear Reviewer gapk,
>
> We would like to remind you that the rebuttal will be closed in **less than 24 hours**. To allow us sufficient time to discuss with you your concerns about our work, we would appreciate it if you could take some time to read our rebuttal and give us some feedback. Thank you very much!

---

> ### Author Response · Authors · 2023-11-23
> **We're looking forward to your feedback**
>
> Dear Reviewer gapk,
>
> We would like to remind you that the rebuttal will be closed in **less than 8 hours**. To allow us a last chance to discuss with you any of your remaining concerns about our work, we would appreciate it if you could take some time to read our rebuttal and give us some feedback. Thank you very much!

---

### Official Review · Reviewer_FFuz · 2023-11-07

**Soundness:** 2 fair
**Presentation:** 1 poor
**Contribution:** 2 fair
**Rating:** 3
**Confidence:** 3

**Summary:**

The authors present a framework for GNN explanation through the len of multilinear extension. To improve the expressiveness they proposed a new method, namely GMT. One variant of GMT is to make sure the GNN is fully linear. Another neural version is to simply re-train a GNN classifier with a frozen subgraph extractor.

**Strengths:**

The paper offered some interesting insights into interpretable GNNs and proposed some solutions. The experiments on the proposed methods are extensive and shown pretty good results.

**Weaknesses:**

First of all, I find the paper hard to read and follow. The presentation and writing need some good work to make the paper more readable and clear. While the authors seem to had a hard time squeeze the paper below the page limit, they also over cite even by a very conservative standard. The citation list is almost the same length as the paper itself.
The experiment set-up is actually quite complicated as detailed in appendix E. The authors also searched extensively for different setups (stated in F2). It is unclear how sensitive the result is, wrt. warm-up strategy and a bunch of sampling strategies in Appendix E2 (page 32).

**Questions:**

Is there any reason or hypothesis that both SubMT (Figure 2) and GMT (Figure 6/7 in Appendix) perform better on Mutag but less well on BA-2Motifs?

---

> ### Author Response · Authors · 2023-11-18
> **Response to Reviewer FFuz [1/3]**
>
> We appreciate your time and efforts in reviewing our paper. Please find our responses below to your concerns.
>
> > W1.1. The presentation and writing need some good work to make the paper more readable and clear.
>
> We are committed to making our paper clearer and would appreciate it if you could help us improve the writing by pointing to us the specific places where we should make it clearer.  For your other comments in W1, we have tried to address some of your concerns in W1 in our responses W1.2 and W1.3 below.
>
> > W1.2. The citation list is almost the same length as the paper itself.
>
> We’d like to clarify that, for a new and crucial topic like the expressivity of interpretable GNNs, it is usually unavoidable to have connections with many other topics and applications. As discussed in our paper, the expressivity of interpretable GNNs has a close relation with faithful interpretation and reliable generalization and has many applications such as interpretable scientific discovery. **We aim to provide an integrated context by citing the most related papers for readers to better understand the position of our work in the literature**.
>
> In fact, we also provide a most concise context in the main paper, and leave the complete version in the appendix for interested readers who are not very familiar with the relevant literature. Our practice of citation follows the convention of the literature such as [1,2,3]. Nevertheless, we are open to any suggestions you feel would improve the readability of our paper.

---

> ### Author Response · Authors · 2023-11-18
> **Response to Reviewer FFuz [2/3]**
>
> > W1.3. The experiment set-up is actually quite complicated as detailed in appendix E.
>
> We’d like to clarify that **we did not extensively search all combinations** of the implementation options as given in Appendix E. We chose to provide the complete list of the implementation choices that we have tried in Appendix E, in order to provide an integrated context to readers of what has been carried out in the research. We believe the discussion of successes and failures in implementation could help readers better understand and improve our work in the future.
>
> In fact, **it does not require an exhaustive search of all combinations** of the implementation options as given in Appendix E, as GMT separates the training into two stages: i) subgraph extractor training; ii) classifier training:
>
> i) The warmup strategies and the hyperparameters in Appendix F.2 are mainly used in training the subgraph extractor. In the experiments, we follow a trial-and-error setting.
> - Since warmup strategies are mainly designed to avoid the trivial solution, it can be inspected with a few runs from the validation performance whether the original training without warmup will stuck in the trivial solutions.
> - We will only use the warmup strategies together (i.e., a larger initial $r$, a larger penalty score, and single subgraph sampling) if we observe a performance degeneration in the validation set. Otherwise, we will stick to the original receipt.
> - In fact, we do not use warmups for most of the setups. Only in BA-2Motifs and MNIST-75sp with GIN, and in Tau3Mu with EGNN, we need the warmups. We have supplemented the details in Appendix E and F to avoid any misunderstandings.
>
> ii) The sampling strategies are mainly used for the training of classifiers based on the subgraph extractor from stage i). **The hyperparameter tuning for classifier training is independent of stage i)**. One could simply select the best subgraph extractor based on the validation performance, and train the classifier onto the selected subgraph extractor. We consider the following 9 decoding strategies from the initialization, architecture, and attention perspectives:
> - Initialization: "new" refers to that the classifier is initialized from scratch; "old" refers to that the classifier is initialized from the subgraph extractor;
> - Architecture: "mul" refers to the default message passing architecture; "lin" refers to the GMT-lin architecture;
> - Attention: "normal" refers to the default weighted message passing scheme; "min0" refers to setting the minimum p% attention scores directly to 0; "max0" refers to setting the maximum p% attention scores directly to 1; "min0max1" refers to setting the maximum p% attention scores directly to 1 while setting the minimum (1-p)% attention scores directly to 0;
>
> |                |              |           | Generalization  |                 |                 | Interpretation  |                 |                 |
> |----------------|--------------|-----------|-----------------|-----------------|-----------------|-----------------|-----------------|-----------------|
> | Initialization | Architecture | Attention | spmotif-0.5     | spmotif-0.7     | spmotif-0.9     | spmotif-0.5     | spmotif-0.7     | spmotif-0.9     |
> |                |              | GSAT      | 47.45+-5.87     | 43.57+-3.05     | 45.39+-5.02     | 74.49+-4.46     | 72.95+-6.40     | 65.25+-4.42     |
> | new            | mul          | min0      | **60.09+-5.57** | 54.34+-4.04     | **55.83+-5.68** | 85.50+-2.40     | **84.67+-2.38** | 73.49+-5.33     |
> | old            | mul          | min0      | 58.83+-7.22     | **55.04+-4.73** | 55.77+-5.97     | **85.52+-2.41** | 84.65+-2.42     | 73.49+-5.33     |
> | new            | mul          | max1      | 44.49+-2.65     | 49.77+-2.31     | 50.22+-2.79     | 85.50+-2.39     | 84.66+-2.37     | 73.50+-5.31     |
> | old            | mul          | max1      | 45.91+-2.86     | 49.11+-3.04     | 50.30+-2.07     | 85.49+-2.39     | 84.64+-2.39     | 73.50+-5.35     |
> | old            | mul          | min0max1  | 51.21+-6.46     | 50.91+-6.50     | 53.13+-4.46     | **85.52+-2.41** | 84.66+-2.43     | 73.49+-5.34     |
> | new            | mul          | normal    | 47.69+-5.72     | 44.12+-5.44     | 40.69+-4.84     | 84.69+-2.40     | 80.08+-5.37     | 73.48+-5.34     |
> | old            | mul          | normal    | 45.36+-2.65     | 44.25+-5.41     | 43.43+-5.44     | 83.52+-3.41     | 80.07+-5.35     | 73.49+-5.36     |
> | new            | lin          | normal    | 43.54+-5.02     | 47.59+-4.78     | 46.53+-3.27     | 85.47+-2.39     | 80.07+-5.37     | **73.52+-5.34** |
> | old            | lin          | normal    | 46.18+-3.03     | 46.42+-5.63     | 49.00+-3.34     | 83.51+-3.39     | 80.09+-5.34     | 73.46+-5.35     |

---

> ### Author Response · Authors · 2023-11-18
> **Response to Reviewer FFuz [3/3]**
>
> The table above shows the generalization and interpretation performance of GMT-sam in the spurious motif datasets, denoted as "spmotif" with different levels of spurious correlations. It can be found that **GMT-sam is generically robust to the different choices of the decoding scheme and leads to improvements with most setups**.
>
>
> > Q1. Why GMT perform less better on BA_2Motifs than Mutag?
>
> First, we’d like to clarify that the SubMT results in Figures 2/6/7 all correspond to GMT-sam implementations.
> Although GMT already achieves two times higher counterfactual fidelity, we note that the empirical performance of GMT can be affected by the optimization of the network. As we have discussed there exists a trivial solution to the graph information bottleneck objective in GSAT, and multi-sampling strategies in GMT can more easily lead the optimization to the trivial solution in simple tasks like BA_2Motifs.
>
> To better demonstrate the theoretical advance of SubMT and avoid the side effects of the optimization, we rerun the counterfactual fidelity experiments with GMT using warmup strategies. As shown in the revised Figures 2, 6, and 7, GMT can achieve even higher counterfactual fidelity than GSAT where GMT achieves 3 times larger counterfactual fidelity compared to GSAT on both BA_2Motifs and Mutag, as GMT approximates the SubMT better.
>
> Please let us know if there are any outstanding questions, and we’d sincerely appreciate it if you could jointly consider our responses above when making a final evaluation of our work.
>
> ## References
>
> [1] How Neural Networks Extrapolate: From Feedforward to Graph Neural Networks, ICLR 2021.
>
> [2] In Search of Lost Domain Generalization, ICLR 2021.
>
> [3] Interpretable Geometric Deep Learning via Learnable Randomness Injection, ICLR 2023.

---

> ### Author Response · Authors · 2023-11-20
> **A summary of response to Reviewer FFuz**
>
> Dear Reviewer FFuz,
>
> We once again appreciate your time and efforts in reviewing our paper. We have revised our manuscript to improve its clarity following your suggestions. In summary,
> - We discussed more clearly for the experimental setup and clarified that the hyperparameter search of GMT-sam has a similar complexity as previous works;
> - Supplementary to our ablation study, we also tested the sensitivity of GMT-sam regarding different subgraph sampling/decoding schemes, and found GMT-sam is generically robust to different;
> - We also discussed the use of warmup strategies more clearly that they mainly helped with escaping from the trivial solution in our experiments. With warmups, GMT-sam could better approximate SubMT and achieve better counterfactual fidelity.
>
> We would appreciate it if you could take a look at our responses and let us know if any of your remaining concerns are not addressed, and we will try our best to address them.

---

> ### Author Response · Authors · 2023-11-21
> **A gentle reminder for the closing rebuttal window**
>
> Dear Reviewer FFuz,
>
> We would like to remind you that the rebuttal will be closed **within 48 hours**. To allow us sufficient time to discuss with you your concerns about our work, we would appreciate it if you could take some time to read our rebuttal and give us some feedback. Thank you very much.

---

> ### Comment · Area_Chair_9YMh · 2023-11-21
>
> Dear reviewer FFuz,
>
> Your negative review has very little substance. If you do not respond to the rebuttal, defend your negative score, and make your criticism more concrete (e.g., where specifically is the readability and clarity of the paper improvable?), I will discard your review when making the accept/reject recommendation.

---

> ### Author Response · Authors · 2023-11-22
> **The rebuttal window will be closed very soon**
>
> Dear Reviewer FFuz,
>
> We would like to remind you that the rebuttal will be closed in **less than 24 hours**. To allow us sufficient time to discuss with you your concerns about our work, we would appreciate it if you could take some time to read our rebuttal and give us some feedback. Thank you very much!

---

> ### Author Response · Authors · 2023-11-23
> **The rebuttal window is closing very soon**
>
> Dear Reviewer FFuz,
>
> We would like to remind you that the rebuttal will be closed in **less than 8 hours**. To allow us a last chance to discuss with you any of your remaining concerns about our work, we would appreciate it if you could take some time to read our rebuttal and give us some feedback. Thank you very much!

---

### Author Response · Authors · 2023-11-18
**Code link**

Dear ACs and Reviewers,

In case you'd be interested, we provide an anonymous link to our sample code for reproducing the results in our paper: https://anonymous.4open.science/r/GMT-C228   .

Best regards,

The 7185 authors.

---

### Author Response · Authors · 2023-11-18
**We have uploaded a revised version of our manuscript**

Dear Reviewers,

Thank you for your time and comments in reviewing our paper. To summarize, all reviewers agree that our results are interesting and insightful, and our empirical studies are extensive and significant. The paper is well-written (pmQk) and well-motivated (gapk). The claims are sound and well supported both theoretically and empirically (gapk, pmQk, uKEu).

We believe the reviewers’ concerns can be addressed and we have revised our draft accordingly. The revised contents are highlighted in blue. In the following, we brief our responses to the main concerns and suggestions raised in the review.

Regarding the hyperparameter search and sensitivity of GMT-sam to different setups, we’d like to clarify that, **the hyperparameter search of two training stages of GMT-sam can be conducted independently**. Therefore the number of overall hyperparameter search space are similar to that of previous works such as GSAT.  We also show empirical results in the [response](https://openreview.net/forum?id=dVq2StlcnY&noteId=eUpuk9kzPO) to Reviewer FFuz that GMT-sam is insensitive to a variety of setups.

We also conducted extensive analysis in response to the questions raised in the review:
- **[Empirical support for proposition 3.3](https://openreview.net/forum?id=dVq2StlcnY&noteId=78bfljP6Pf)** show that proposition 3.3 is well supported;
- **[Results with additional interpretation baselines based on PNA](https://openreview.net/forum?id=dVq2StlcnY&noteId=78bfljP6Pf)** show that the advance of GMT-sam remain siginifcant and consisten;
- **[Time complexity analysis](https://openreview.net/forum?id=dVq2StlcnY&noteId=7Dt6KFXS74)** show that the training of GMT-sam will not cost much time empirically, while GMT-sam enjoys a similar inference latency as GMT-lin and GSAT;
- **[The counterfactual fidelity measured in different distance metrics](https://openreview.net/forum?id=dVq2StlcnY&noteId=9Dk54pCvMA)** show that GMT-sam can lead to even 10 times larger improvements than GSAT when measured via KL divergence or JSD divergence.

Besides, we have revised our manuscript and improved its clarity thanks to the suggestions of all reviewers.

We hope our responses could address the concerns of all reviewers. And we are happy to discuss and provide further evidence for any outstanding questions. We would appreciate it if you could take our responses into consideration when making the final evaluation of our work.

Sincerely,

Authors

---

### Meta-Review · Area_Chair_9YMh · 2023-12-15

**Metareview:**

This work addresses the limitations of existing interpretable graph neural networks (XGNNs) in learning accurate representations of interpretable subgraphs. While previous approaches predominantly use attention-based mechanisms to learn edge or node importance, the representational properties and constraints of such methods have not been thoroughly investigated. The paper introduces a theoretical framework, Subgraph Multilinear Extension (SubMT), that formulates interpretable subgraph learning through the multilinear extension of the subgraph distribution.

Multiple comparisons with GSAT throughout the paper suggest that the primary distinction lies in the sampling process, with the proposed method employing a higher number of samples. It would be beneficial for the authors to emphasize this key differentiator and provide a more detailed discussion on the impact of increased sampling on method performance, offering insights into the advantages of GSAT.

While one of the reviewers was positive about the paper, assigning a score of 8, during the discussion after the rebuttal phase they emphasized their limited understanding of the work underlined with a confidence score of 2. Hence, after reading all reviews and responses, and given the overall score, I tend towards rejecting the paper. However, I will also indicate that a bump-up is possible.

**Justification For Why Not Higher Score:**

Several weaknesses pointed out by the reviewers.

**Justification For Why Not Lower Score:**

-

---

### Decision · Program_Chairs · 2024-01-16

Reject